# Emerging symmetric strain response and weakening nematic fluctuations in strongly hole-doped iron-based superconductors

P. Wiecki [1], M. Frachet[1], A.-A. Haghighirad [1], T. Wolf[1], C. Meingast[1], R. Heid [1] & A. E. Böhmer[1,2 ✉]

Electronic nematicity is often found in unconventional superconductors, suggesting its relevance for electronic pairing. In the strongly hole-doped iron-based superconductors, the symmetry channel and strength of the nematic fluctuations, as well as the possible presence of long-range nematic order, remain controversial. Here, we address these questions using transport measurements under elastic strain. By decomposing the strain response into the appropriate symmetry channels, we demonstrate the emergence of a giant in-plane symmetric contribution, associated with the growth of both strong electronic correlations and the sensitivity of these correlations to strain. We find weakened remnants of the nematic fluctuations that are present at optimal doping, but no change in the symmetry channel of nematic fluctuations with hole doping. Furthermore, we find no indication of a nematic-ordered state in the $AFe_2As_2$ (A = K, Rb, Cs) superconductors. These results revise the current understanding of nematicity in hole-doped iron-based superconductors.

[1] Karlsruhe Institute of Technology, Institute for Quantum Materials and Technologies, Karlsruhe, Germany. [2] Institut für Experimentalphysik IV, Ruhr-Universität Bochum, Bochum, Germany. ✉email: boehmer@physik.ruhr-uni-bochum.de

Nematicity, the breaking of rotational symmetry by electronic interactions, has by now been observed in a variety of unconventional superconductors. In addition to iron-based superconductors with almost ubiquitous nematicity[1,2], nematicity is discussed in the context of cuprate high-$T_c$ superconductors[3–12], heavy-fermion superconductors[13,14], intercalated $Bi_2Se_3$ topological superconductors[15–18], and even twisted bilayer graphene[19]. Furthermore, it has been theoretically suggested that nematic fluctuations may enhance pairing and therefore be an important ingredient for high-$T_c$ superconductivity[20–23].

In iron-based superconductors, nematicity has been intensively studied in the vicinity of the parent compound $BaFe_2As_2$ because the stripe-type antiferromagnetic ground state inherently breaks the $C_4$ rotational symmetry of the high-temperature tetragonal phase[1,2,24,25], corresponding to a nematic degree of freedom. The accompanying structural distortion is in the $B_{2g}$ channel of the tetragonal $D_{4h}$ point group. In electron-doped $BaFe_2As_2$, the structural distortion occurs at a higher temperature than the antiferromagnetic state, creating a long-range, nematic-ordered phase. Nematic fluctuations of $B_{2g}$ symmetry are observed near optimal doping in both hole- and electron-doped $BaFe_2As_2$[26,27]. Such fluctuations have frequently been studied using elastoresistance, the strain dependence of electrical resistivity[1,28,29].

The parent compound $BaFe_2As_2$ nominally has a $3d^6$ Fe configuration. With hole doping the Fe electron configuration begins to approach the half-filled $3d^5$, where a Mott insulating state is expected theoretically[30–32]. Indeed, signatures of strong electronic correlations and orbital-selective Mott behavior have been observed in the isoelectronic $3d^{5.5}$ series $AFe_2As_2$ (A = K, Rb, Cs), including an enhanced Sommerfeld coefficient and signs of a coherence–incoherence crossover[33–35]. On the basis of the strong increase of electronic correlations with increasing alkali ion size in $AFe_2As_2$ (A = K, Rb, Cs), it has further been proposed that these compounds lie near a quantum critical point (QCP) associated with the suppression of a (thus far unobserved) ordered phase, possibly related to the $3d^5$ Mott insulator[36,37]. Furthermore, the electronic correlations in $AFe_2As_2$ have been found to be highly sensitive to in-plane strain[34,36].

The fate of nematicity in the strongly hole-doped iron-based superconductors remains controversial. In the $Ba_{1−x}K_xFe_2As_2$ series, the elastic softening associated with the $B_{2g}$ nematic fluctuations decreases with hole doping and is no longer observed for $x \geq 0.82$[27]. Several recent studies have suggested a change to nematic fluctuations of $B_{1g}$ symmetry in the $3d^{5.5}$ compounds, in contrast to the pervasive $B_{2g}$ nematic fluctuations observed at optimal doping[38–44]. Furthermore, an ordered $B_{1g}$ nematic phase has been proposed in $RbFe_2As_2$ based on a maximum in the elastoresistance[39] and an anomaly in the zero-field nuclear spin lattice relaxation rate at a similar temperature[44]. In addition, asymmetries were observed in STM at low temperature[40]. However, the study of nematic fluctuations in these compounds is complicated by the emergence of the strong electronic correlations, which are sensitive to in-plane symmetric $A_{1g}$ strain[45]. In such a case, it is essential to properly decompose the contributions in different symmetry channels[45–47]. This symmetry decomposition was not performed in previous elastoresistance measurements on the strongly hole-doped iron-based superconductors[39]. In addition, the extreme thermal expansion of these samples (Fig. 1b) poses an experimental challenge for controlled measurements under elastic strain[45].

In this work, we present comprehensive elastoresistance measurements on the hole-doped iron-based superconductors $Ba_{0.4}K_{0.6}Fe_2As_2$ and $AFe_2As_2$ (A = K, Rb, Cs) carried out in a piezoelectric-based strain cell capable of full control over the strain state of the sample. We find a monotonic increase of the $A_{1g}$ response with hole doping and increasing alkali ion size. In contrast, the $B_{2g}$ elastoresistance, a measure of the $B_{2g}$ nematic susceptibility, weakens with hole doping. The $B_{1g}$ elastoresistance shows no sign of divergence, and there is no indication of the emergence of a $B_{1g}$-type nematic instability with increasing hole doping. Furthermore, we find a clear correspondence between the low-temperature elastoresistance and the electronic Grüneisen parameter if the elastoresistance coefficient is defined based only on the temperature-dependent part of the resistance.

## Results

**Elastoresistance measurements**. The freestanding electrical resistance of our samples is displayed in Fig. 1a. For elastoresistance measurements, the sample is mounted in a strain cell composed of titanium (Fig. 1d). Figure 1b displays the length of the samples as a function of temperature, along with titanium for comparison. We see that the hole-doped samples shrink much more than the titanium apparatus on cooling. Thus, the samples can be maintained near neutral strain on cooling by adjusting the distance between the sample mounting plates in the strain cell based on the thermal expansion difference between the sample and titanium (see "Methods"). We place eight electrical contacts on the sample so that both longitudinal and transverse elastoresistance can be measured in the same experimental run (Fig. 1d). Raw elastoresistance data on $RbFe_2As_2$ obtained in this way are shown in Fig. 1e and f as an example.

In Fig. 2a–h, we present the longitudinal and transverse elastoresistance for all samples. Here, the elastoresistance is defined, for the moment, simply as $m_{ii,xx} = (1/R_{ii})dR_{ii}/d\epsilon_{xx}$, where $i = x$ corresponds to the longitudinal and $i = y$ corresponds to the transverse elastoresistance. The strain is applied along $x \| [100]$ (Fig. 2a–d) or $x \| [110]$ (Fig. 2e–h). In one configuration, we have reproduced our result by the usual method[28] of gluing the sample directly onto a piezoelectric stack (Fig. 2c, inset). These results are in conflict with previous measurements[39]. When strain is applied along [100], the transverse elastoresistance is larger than the longitudinal over most of the temperature range for all samples. In contrast, for strain applied along [110], we observe that the longitudinal and transverse elastoresistance are equal at high temperature, while the longitudinal becomes larger at low temperature, for all samples. In the case of $Ba_{0.4}K_{0.6}Fe_2As_2$ with strain along [110] (Fig. 2e), the longitudinal and transverse elastoresistance have opposite sign. This is the expected behavior for dominant $B_{2g}$ nematic fluctuations[46]. In contrast, in the $3d^{5.5}$ superconductors $AFe_2As_2$ (A = K, Rb, Cs), both the longitudinal and transverse elastoresistance are positive at all temperatures.

We note significant drops in the elastoresistance on decreasing temperature below ~15 K for both transverse and longitudinal, most notably in $KFe_2As_2$ and $RbFe_2As_2$ (Fig. 2b, c, f, g). We do not associate these anomalies with a phase transition, since no hint of a phase transition is seen in either $R(T)$ or the thermal expansion $L(T)$ (Fig. 1), nor indeed in the respective temperature-derivatives $dR(T)/dT$ and $dL(T)/dT$ (Supplementary Fig. 3). Rather, as explained below, the drop in elastoresistance results from the sample's resistance becoming comparable to its residual resistance. Note, however, that the broad peak in $CsFe_2As_2$ at ~35 K is a pronounced manifestation of the coherence–incoherence crossover in this material.

**Physical mechanism for strain dependence of resistance**. To obtain a better physical understanding of elastoresistance, it is useful to take a complementary view and study the resistance as a function of temperature at a fixed strain. This is shown in Fig. 3a, taking $KFe_2As_2$ as an example. At low temperatures, the data can be fit to the standard Fermi liquid form $R = R_0 + AT^2$. The

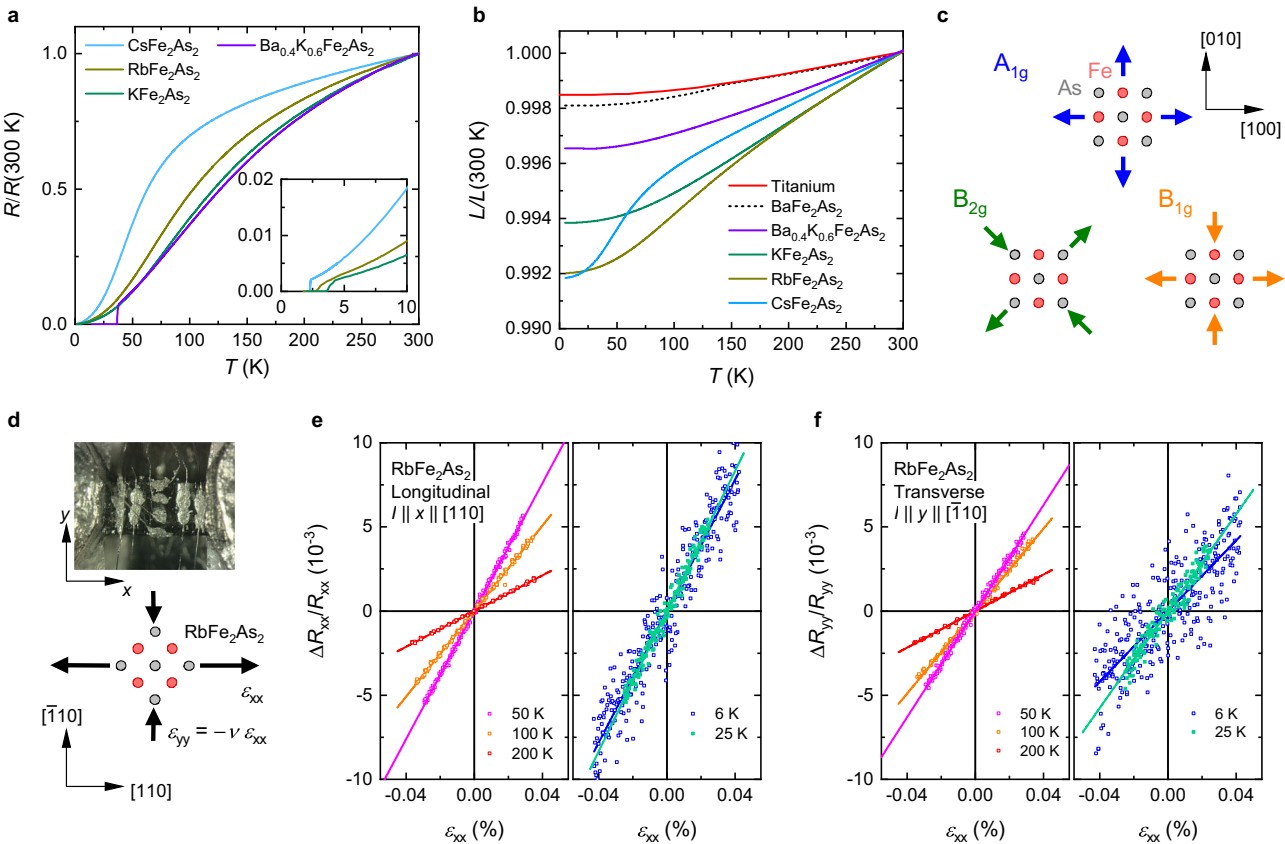

**Fig. 1 Basic material properties and experimental setup. a** Normalized electrical resistance as a function of temperature. Inset: zoom of superconducting transitions. **b** In-plane sample length changes as a function of temperature[34]. Titanium and BaFe$_2$As$_2$[51] are shown for comparison. **c** Sketch of the deformations in the Fe plane corresponding to the indicated irreducible representations. **d** Photograph of a RbFe$_2$As$_2$ sample mounted between titanium plates in the strain cell. The uniaxial stress axis is defined to be $x$. In this example, strain is applied along [110] and the sample experiences $A_{1g} + B_{2g}$ strain. $\nu$ is the in-plane Poisson ratio of the sample. **e**, **f** Longitudinal and transverse elastoresistance curves at indicated temperatures. Lines are linear fits.

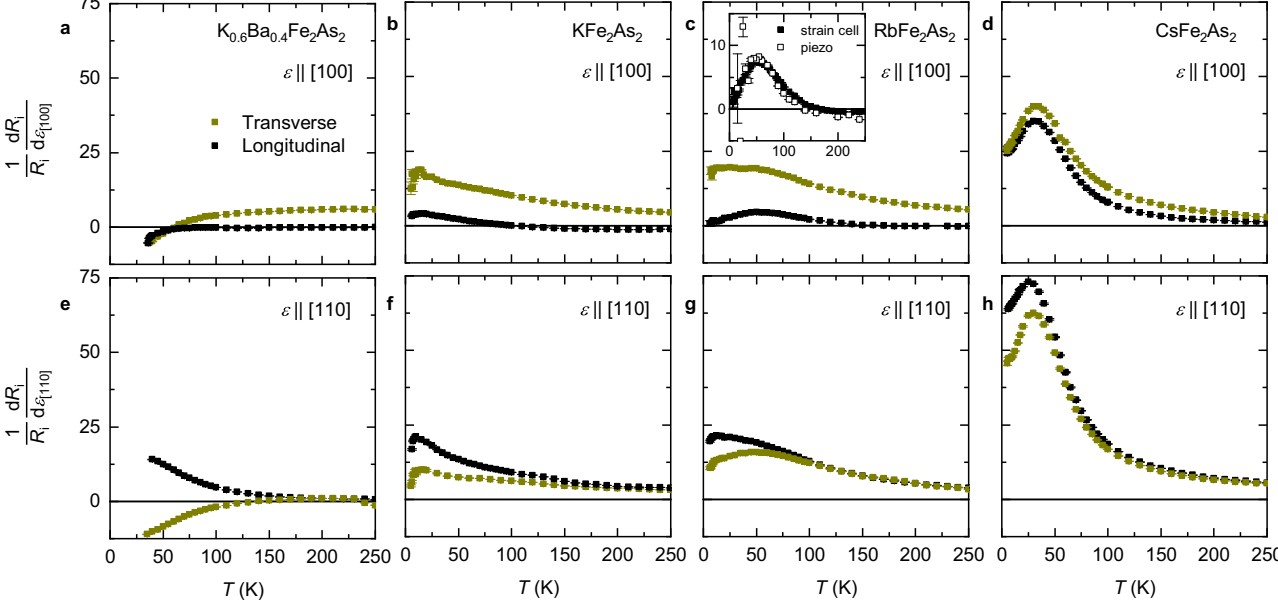

**Fig. 2 Elastoresistance of strongly hole-doped iron-based superconductors.** Longitudinal and transverse elastoresistance with the strain direction $\epsilon_{xx}\|[100]$ (**a–d**) and $\epsilon_{xx}\|[110]$ (**e–h**) for indicated samples arranged by column. Error bars indicate one standard deviation of a linear fit. The inset of **c** shows a comparison of strain-cell-based and piezoelectric stack-based longitudinal elastoresistivity measurements for RbFe$_2$As$_2$ with $\epsilon_{xx}\|[100]$. The strain-cell data have been scaled to account for strain relaxation in the epoxy (see "Methods" and Supplementary Fig. 1).

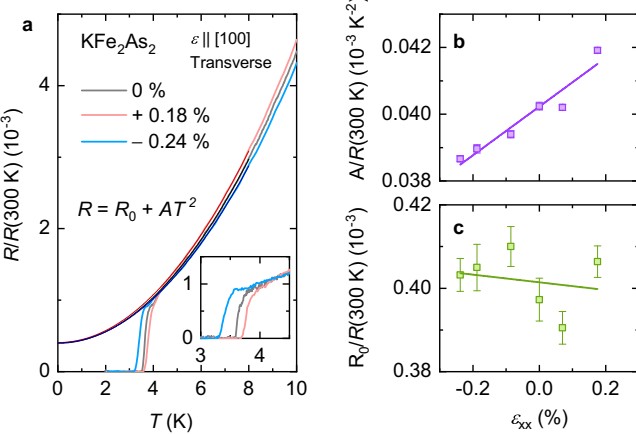

**Fig. 3 Resistance at fixed strain. a** Normalized resistance of $KFe_2As_2$ as a function of temperature at fixed strains, along with fits to the indicated fitting function. Inset: zoom of superconducting transitions. **b** Strain dependence of the fit parameter $A$, measuring electronic correlations and effective mass. **c** Strain dependence of the fit parameter $R_0$, the residual resistance. Lines in **b**, **c** are linear fits. Error bars represent the standard error of the fits.

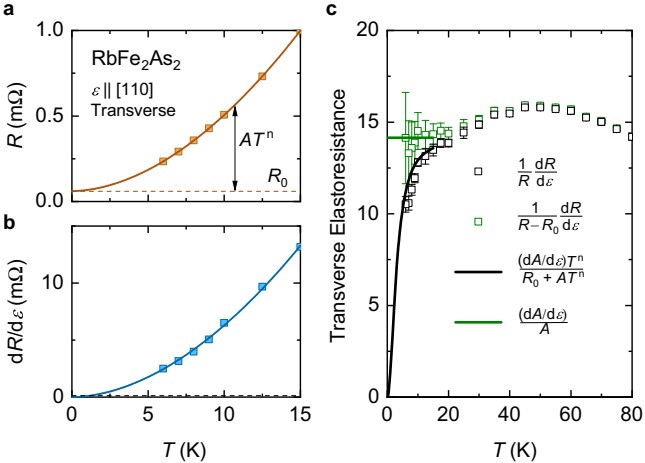

**Fig. 4 Redefinition of of elastoresistance based on the temperature-dependent resistance contribution. a** The resistance of a particular $RbFe_2As_2$ crystal at temperatures where elastoresistance was measured. The solid line is a fit of these points to $R = R_0 + AT^n$ (here $n = 1.84$). The dashed line represents the residual resistance $R_0$. **b** The strain derivative of the sample resistance at selected temperatures. The solid line is a fit of these points to $dR/d\epsilon = B + CT^n$ (again $n = 1.84$). $B$ is consistent with zero within uncertainty. The dashed line gives an upper limit on $B$, from the fit uncertainty. **c** Elastoresistance calculated as $m = (1/R)dR/d\epsilon$ (black) and $\bar{m} = [1/(R - R_0)]dR/d\epsilon$ (green). The error bars on $[1/(R - R_0)]dR/d\epsilon$ reflect the uncertainty in the estimation of $R_0$. The solid lines show the same quantities instead calculated from the fitted functions in (**a**, **b**).

coefficient $A$ is a measure of electronic correlations and effective mass, as given by the Kadowaki–Woods relation for a Fermi liquid $A \propto \gamma^2$, where $\gamma$ is the Sommerfeld coefficient. The fit parameters $R_0$ and $A$ are shown as a function of strain in Fig. 3b and c. We observe that the coefficient $A$ increases linearly with strain, while the residual resistance $R_0$ is strain independent within experimental uncertainty. We have similarly confirmed that $R_0$ is also strain-independent in $CsFe_2As_2$.

From the Kadowaki–Woods relation, the increase of $A$ with strain is quantitatively consistent with the large positive $d\gamma/d\epsilon$

inferred from thermal expansion measurements[34,36] (see Supplementary Note 2) and LDA+DMFT calculations[48]. Consistently, hydrostatic pressure reduces $A$ in $KFe_2As_2$[49]. Furthermore, we observe that $T_c$ increases with tensile strain. The strain dependence of $T_c$ is in rough quantitative agreement with inferences based on thermodynamic measurements[36] (see "Methods").

In $RbFe_2As_2$ and $CsFe_2As_2$, we find that the resistance at low temperature can be fit to $R = R_0 + AT^n$, but with an exponent $n < 2$ (Fig. 4a). This is consistent with the non-Fermi liquid behavior seen at low temperature in these samples by NMR[37]. In Fig. 4b, we show the derivative $dR/d\epsilon$ as a function of temperature for the case of $RbFe_2As_2$ with strain ∥[110] as a representative example. Taking $R_0$ to be independent of strain, $dR/d\epsilon = (dA/d\epsilon)T^n$. Consistently, we find that $dR/d\epsilon$ can be fit to $B + CT^n$ with the same exponent $n$ as the resistance and an intercept $B$ consistent with zero. In this situation, if the elastoresistance is simply defined as $m = (1/R)dR/d\epsilon$ as is commonly done (see Fig. 2), we obtain $m = (dA/d\epsilon)T^n/(R_0 + AT^n)$. Clearly, $m \to 0$ as $T \to 0$, as shown in Fig. 4c. As a consequence, the elastoresistance starts to drop on cooling when the residual resistance becomes a significant fraction of the total resistance below 15 K, as noted earlier (Fig. 2). If instead the elastoresistance is redefined based on only the temperature-dependent contribution to the resistance as $\bar{m} = [1/(R - R_0)]dR/d\epsilon$, we obtain $\bar{m} \to (dA/d\epsilon)/A$ at low temperature (Fig. 4c). Note that these arguments are valid for any $n$. A similar approach has been applied in the analysis of elastoresistance data in $YbRu_2Ge_2$[50].

The redefined elastoresistance $\bar{m}$ produces more physically meaningful results at low temperature. For example, in the special case of $n = 2$ where $A \propto \gamma^2$, $\bar{m}(T \to 0)$ is given by $(dA/d\epsilon)/A = (d\gamma^2/d\epsilon)/\gamma^2 = 2(d\gamma/d\epsilon)/\gamma$, which is proportional to the Grüneisen parameter $\Gamma \equiv (d\gamma/d\epsilon)/\gamma$ for the Sommerfeld coefficient $\gamma$. $\Gamma$ is a measure of the strain dependence of an energy scale. Its divergence is a central characteristic of a strain-tuned QCP[51–53], as previously discussed in these materials[36].

Having understood the longitudinal and transverse elastoresistance data, we now calculate the symmetry-decomposed elastoresistance coefficients $\bar{m}_\alpha$, where $\alpha$ represents the irreducible representation of the tetragonal $D_{4h}$ point group. These coefficients determine the resistance changes associated with the pure symmetric strains illustrated schematically in Fig. 1c. In terms of the experimental data, $\bar{m}_{A_{1g}}$ is proportional to the sum of the longitudinal and transverse elastoresistance, while $\bar{m}_{B_{1g}}$ ($\bar{m}_{B_{2g}}$) is proportional to the difference when $\epsilon\|[100]$ ($\epsilon\|[110]$) (see "Methods"). The resulting symmetry-decomposed elastoresistance coefficients are shown in Fig. 5. The in-plane symmetric coefficient $\bar{m}_{A_{1g}}$ (Fig. 5a) strongly increases in magnitude with hole doping from $Ba_{0.4}K_{0.6}Fe_2As_2$ to $KFe_2As_2$ and further with chemical substitution from $KFe_2As_2$ to $CsFe_2As_2$. Furthermore, $\bar{m}_{A_{1g}}$ of $CsFe_2As_2$ has strong temperature dependence, with an increase on cooling and a broad peak near 30 K. This temperature dependence is reminiscent of the coherence–incoherence crossover observed in the thermal expansion coefficient of this material[34,45]. The coefficients $\bar{m}_{B_{2g}}$ (Fig. 5b) and $\bar{m}_{B_{1g}}$ (Fig. 5c) are significantly smaller in magnitude than $\bar{m}_{A_{1g}}$, except for the case of $\bar{m}_{B_{2g}}$ in $Ba_{0.4}K_{0.6}Fe_2As_2$. For all samples, $\bar{m}_{B_{2g}}$ is near zero at high temperature and displays a divergent increase on cooling. The positive sign of $\bar{m}_{B_{2g}}$ is opposite to that of $BaFe_2As_2$, consistent with the observed sign change of resistivity anisotropy upon K substitution in $Ba_{1-x}K_xFe_2As_2$[54]. We also note that both $\bar{m}_{B_{1g}}$ and $\bar{m}_{B_{2g}}$ of $Ba_{0.6}K_{0.4}Fe_2As_2$ are similar in magnitude and sign to $CaKFe_4As_4$, which is isoelectronic to $Ba_{0.5}K_{0.5}Fe_2As_2$[55,56].

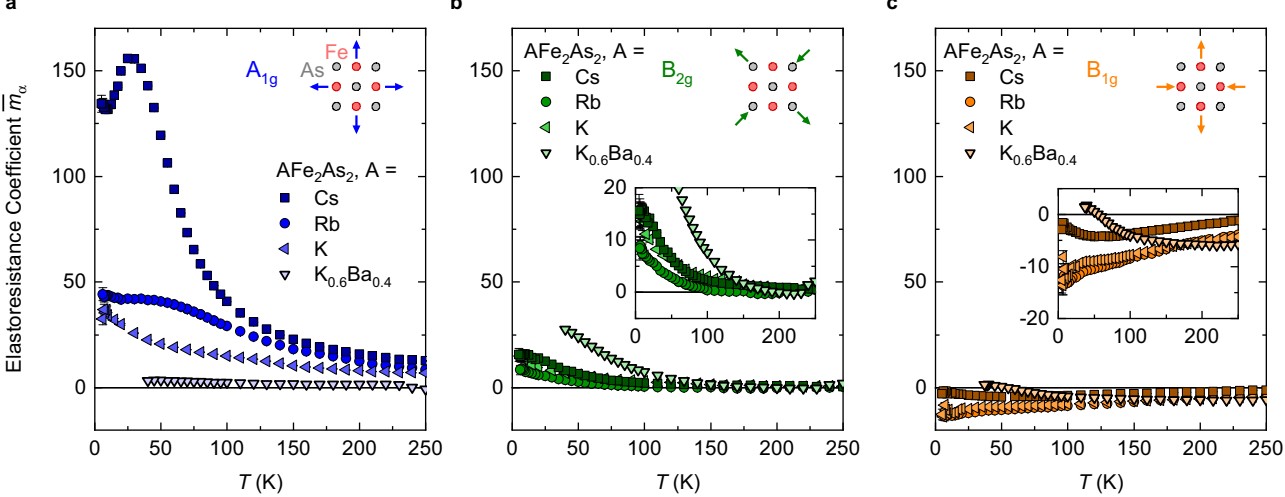

**Fig. 5 Symmetry-decomposed elastoresistance coefficients.** Calculated with the redefined $\bar{m}_\alpha$ (see text). **a** $\bar{m}_{A_{1g}}$ calculated from $\epsilon_{xx} \| [110]$ data. **b** $\bar{m}_{B_{2g}}$. **c** $\bar{m}_{B_{1g}}$. The strain symmetry channels are illustrated in the upper right of each panel. The insets in panels **b** and **c** show a zoomed view. The error bars take into account the uncertainty in $R_0$. Due to large uncertainties in $R_0$ for the high-$T_c$ sample Ba$_{0.4}$K$_{0.6}$, we have plotted $m_\alpha$ instead of $\bar{m}_\alpha$ for this sample (points with thick edges).

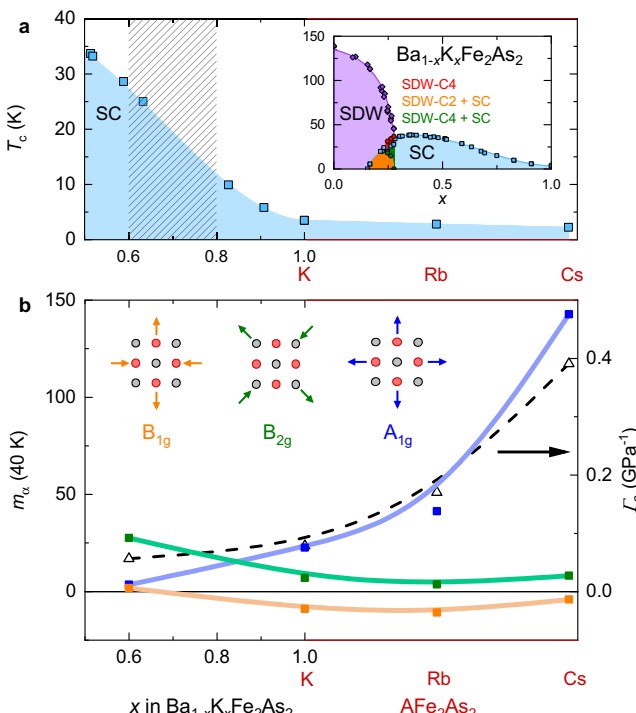

**Fig. 6 Evolution of the elastoresistance coefficients with chemical substitution. a** Phase diagram of the hole-doped series Ba$_{1-x}$K$_x$Fe$_2$As$_2$ for $x$ > 0.5 and the isoelectronic series $3d^{5.5}$ series AFe$_2$As$_2$ (A = K, Rb, Cs) on the same horizontal axis. The hashed area denotes the doping range of a Lifshitz transition[58–61] and proposed broken time-reversal symmetry in the SC state[62–64]. *XY*-type nematic fluctuations were suggested in the same doping range in the related system Ba$_{1-x}$Rb$_x$Fe$_2$As$_2$[39]. Inset: full thermodynamic phase diagram of Ba$_{1-x}$K$_x$Fe$_2$As$_2$. **b** Value of the elastoresistance coefficients $\bar{m}_\alpha$ at 40 K as a function of doping, for the three symmetry channels. The black triangles (right axis) represent the in-plane electronic Grüneisen parameter $\Gamma$, for comparison with $\bar{m}_{A_{1g}}$. Note the experimental definition[34,51] $\Gamma_a \equiv \alpha_a^{\text{elec}}/C^{\text{elec}} = -(d\gamma/dp_a)/\gamma$ where $p_a$ is the in-plane uniaxial pressure.

The coefficient $\bar{m}_{B_{1g}}$ is nonzero at high temperature and decreases roughly linearly on cooling in RbFe$_2$As$_2$ and KFe$_2$As$_2$. In CsFe$_2$As$_2$ and Ba$_{0.4}$K$_{0.6}$Fe$_2$As$_2$, we observe an upturn at low temperature. The temperature dependence of $\bar{m}_{B_{1g}}$ shows no sign of divergence in any of our samples.

## Discussion

We discuss first the in-plane symmetric coefficient $\bar{m}_{A_{1g}}$ (Fig. 5a). Since the resistance reflects the electronic entropy and correlations in these materials[35,45], the increase of $\bar{m}_{A_{1g}}$ from KFe$_2$As$_2$ to CsFe$_2$As$_2$ indicates that the strain sensitivity of the electronic correlations increases as a result of this chemical substitution. This is consistent with the increase in the strain derivative of the Sommerfeld coefficient $\partial\gamma/\partial\epsilon$ observed in thermodynamic measurements[34,36]. Therefore, the origin of the large $A_{1g}$ elastoresistance is the high-strain sensitivity of the electron correlations associated with the orbital-selective Mott behavior. As a consequence, the temperature and substitution dependence of the $A_{1g}$ coefficient show a strong similarity to that of the thermal expansion coefficient[34,45] (Supplementary Note 3). We note that the heavy-fermion superconductor URu$_2$Si$_2$ has also recently been shown to have a large symmetric elastoresistance[57].

We turn now to the coefficients $\bar{m}_{B_{2g}}$ (Fig. 5b) and $\bar{m}_{B_{1g}}$ (Fig. 5c) corresponding to symmetry-breaking strain. They are proportional to the nematic susceptibility in the respective symmetry channels[1], though the proportionality constant may become temperature and material dependent[55]. Note that the underdoped BaFe$_2$As$_2$ compounds are characterized by a diverging $\bar{m}_{B_{2g}}$. For all samples, $\bar{m}_{B_{2g}}$ is near zero at high temperature and displays a divergent increase on cooling. In AFe$_2$As$_2$ (A = K, Rb, Cs), $\bar{m}_{B_{2g}}$ is smaller than in Ba$_{0.4}$K$_{0.6}$Fe$_2$As$_2$ but has a similar temperature dependence, strongly suggesting that we are observing the remnants of the $B_{2g}$ nematic fluctuations found in lightly doped BaFe$_2$As$_2$. The temperature dependence of the coefficient $\bar{m}_{B_{1g}}$ does not show a divergence with decreasing temperature, an indication that the AFe$_2$As$_2$ (A = K, Rb, Cs) compounds are not near a $B_{1g}$ nematic instability. The evolution of $\bar{m}_{B_{1g}}$ and $\bar{m}_{B_{2g}}$ show no indication of a change from $B_{2g}$ to $B_{1g}$ nematic

fluctuations in hole-doped iron-based superconductors, in contrast to previous studies[39–43]. Further, nematic fluctuations of XY, as opposed to Ising, character have been proposed in the $Ba_{1-x}Rb_xFe_2As_2$ series in the $0.6 < x < 0.8$ doping range[39]. Whereas we do find that $\bar{m}_{B_{1g}}$ and $\bar{m}_{B_{2g}}$ have similar magnitude in this doping range (in $Ba_{0.4}K_{0.6}Fe_2As_2$ in our case), their distinct temperature dependence is inconsistent with XY nematic fluctuations.

A phase transition to a long-range ordered $B_{1g}$ nematic-ordered phase below 40 K has been proposed in $RbFe_2As_2$ based on a maximum in the longitudinal elastoresistance measured along [100][39]. The symmetry decomposition of the elastoresistance reveals that such a maximum occurs in the $A_{1g}$ channel, which we associate with the coherence/incoherence crossover and not a phase transition. Consistently, the resistance and thermal expansion[34] data show no sign of a phase transition in $RbFe_2As_2$ (Fig. 1a and Supplementary Fig. 3).

Our results are summarized in Fig. 6. In this phase diagram, we plot the values of the elastoresistance coefficients at 40 K as a function of substitution, with both the hole-doped series $Ba_{1-x}K_xFe_2As_2$ and the isoelectronic $3d^{5.5}$ series $AFe_2As_2$ (A = K, Rb, Cs) on the same horizontal axis. In all, 40 K is chosen so that the high-$T_c$ sample can be included. The hashed region $0.6 < x < 0.8$ denotes the doping range where a significant change of behavior has been observed, including a Lifshitz transition[58–61] and proposed broken time-reversal symmetry in the superconducting state[62–64], as well as the proposed XY nematic state in the Rb-doped series ($Ba_{1-x}Rb_xFe_2As_2$)[39]. Our results suggest that this doping range coincides with the emergence of an enhanced $A_{1g}$ elastoresistance. Recalling that $\bar{m}_{A_{1g}}$ is expected to reflect the in-plane Grüneisen parameter $\Gamma_a$ at low temperature, we plot $\Gamma_a$ alongside $\bar{m}_{A_{1g}}$ (40 K) in Fig. 6. The agreement confirms that the $\bar{m}_{A_{1g}}$ component of the elastoresistance is associated with the strong electronic correlations observed in thermodynamic measurements.

We stress that we observe, nevertheless, the remnant of the $B_{2g}$ nematic fluctuations of the optimally doped regime even in fully substituted samples with $3d^{5.5}$ electronic configuration. We also observe signatures of possible weak nematic fluctuations in the $B_{1g}$ channel, but we do not observe a change from $B_{2g}$-dominant to $B_{1g}$-dominant nematic fluctuations with doping and we find no evidence for a bulk nematic state in any of our samples. These results revise the current understanding of nematicity in hole-doped iron-based superconductors and raise some new points. First, neither $B_{1g}$ nor $B_{2g}$ nematic fluctuations depend strongly on the alkali ion size in $AFe_2As_2$ (A = K, Rb, Cs). In contrast, the strong correlations related to the orbital-selective Mott behavior seen in the $A_{1g}$ channel increase dramatically with increasing alkali ion size. The contrasting behavior indicates that the orbital-selective Mott physics does not cause the nematic fluctuations. Second, we find that the $B_{2g}$ nematic fluctuations are surprisingly robust, persisting at high hole doping even beyond the Lifshitz transition where the electron pocket of the Fermi surface and the corresponding nesting properties are lost, and magnetic fluctuations become incommensurate.

To conclude, we reflect on the different physical mechanisms responsible for the observed strain-induced resistance changes. In $Ba_{0.4}K_{0.6}Fe_2As_2$ with a sizeable nematic susceptibility, anisotropic strains change the measured resistance by favoring the resistance of one in-plane direction over the other, as nematic order causes a resistance anisotropy. In $CsFe_2As_2$, by contrast, symmetric strains directly modulate the strength of electron correlations and effective mass causing changes in average in-plane resistance.

## Methods

**Sample growth**. Single crystals of $CsFe_2As_2$ were grown from solution using a Cs-rich self-flux in a sealed environment[65]. Cs, Fe, and As were weighted in molar ratio 8:1:11, respectively. All sample manipulations were performed in an argon glovebox ($O_2$ content is < 0.5 ppm). Molten Cs together with a mixture of iron and arsenic powder were loaded into an alumina crucible. Typically 15–20 g of a mixture of Cs, Fe, and As were used for each crystal growth experiment. The alumina crucible with a lid was placed inside a stainless steel container and encapsulated. This was done by welding a stainless screw cap to one end of the container. This has the advantage that stainless steel does not corrode due to Cs vapor at high temperatures. The stainless steel container was placed in a tube furnace, which was evacuated at 5–10 mbar and slowly heated up to 200 °C. The sample was kept at this temperature for 10 h and subsequently heated up to 980 °C in 8 h. The furnace temperature was kept constant at 980 °C for 5 h and slowly cooled to 760 °C in 14 days, and then the furnace was canted to separate the excess flux. After cooling to room temperature, shiny plate-like crystals were easily removed from the remaining ingot. Refined crystallographic data have been presented elsewhere[45]. $KFe_2As_2$ and $RbFe_2As_2$ single crystals were obtained under similar conditions using an alkali metal/As-rich flux[34].

High-quality single crystals of $Ba_{0.4}K_{0.6}Fe_2As_2$ were grown by a self-flux technique, using FeAs fluxes, in alumina crucibles sealed in iron cylinders using very slow cooling rates of 0.2–0.4 °C/h. The crystals were annealed in situ by further slow cooling to room temperature.

**Elastoresistance measurements**. Elastoresistance measurements were performed in a commercial strain cell (Razorbill Instruments CS-120). The samples were cut with edges along the tetragonal in-plane directions [100] or [110], with typical dimensions of $3.0 \times 1.0 \times 0.06$ mm. The samples were fixed in the strain cell using DevCon 5 Minute Epoxy with an effective strained sample length of 2.0 mm. The epoxy thickness was controlled to be 50 μm below the sample and ~30 μm above the sample. Cigarette paper was used to ensure the electrical insulation of the sample from the titanium mounting plates. The sample strain was measured via a built-in capacitive displacement sensor and readout on an Andeen Hagerling 2550A capacitance bridge. Resistance was measured using a four-point method on a Lake Shore 372 resistance bridge. Hans Wolbring Leitsilber 200N was used to make the electrical contacts, as contacts made with DuPont 4929N silver paint were found to be mechanically unstable on strain application.

We note that the true strain experienced by a sample in the strain cell is less than the nominal strain read by the capacitance sensor[66]. From a comparison with established elastoresistance data on $BaFe_2As_2$, we estimate that the true strain is ~60% of the nominal strain in our setup, as shown in Supplementary Fig. 1. As all measurements are performed with very similar sample dimensions and sample mounting in the same cell, conversion of the elastoresistance from nominal to true strain can be performed by scaling them up by a factor of (1/0.6). We report primarily nominal strain, the only exception being the inset of Fig. 2c where the scaling factor has been applied in order to compare with piezoelectric stack-based measurements.

Due to the large thermal expansion mismatch between our samples and the titanium body of the strain cell, the samples will experience tension on cooling (Fig. 1b). In the most extreme case of $CsFe_2As_2$, the sample is expected to experience ~0.7% tension at base temperature. In contrast, the parent compound $BaFe_2As_2$ has a thermal expansion nearly identical to titanium. This is also true for lightly doped compounds[66]. Elastoresistance has typically been measured by gluing the sample directly onto the side of a piezoelectric stack, which has a small thermal expansion, similar to titanium. The success of previous elastoresistance measurements on lightly doped $BaFe_2As_2$ samples thus depended on the fortunately similar thermal expansion of the piezoelectric stack and the sample. However, this method is unreliable for the $3d^{5.5}$ superconductors[45].

To keep the sample in an unstrained state on cooling, we progressively reduce the distance between the sample mounting plates by an amount calculated based on the known thermal expansion difference between the sample and titanium, and the measurement of a titanium calibration sample[45].

To express the elastoresistance in terms of the irreducible representations $\alpha$ of the $D_{4h}$ point group, we note that the symmetric strains $\epsilon_\alpha$ are given in terms of $\epsilon_{xx}$ by $\varepsilon_{A_{1g}} = \frac{1}{2}(\varepsilon_{[100]} + \varepsilon_{[010]}) = \frac{1}{2}(\varepsilon_{[110]} + \varepsilon_{[\bar{1}10]})$, $\varepsilon_{B_{1g}} = \frac{1}{2}(\varepsilon_{[100]} - \varepsilon_{[010]})$ and $\varepsilon_{B_{2g}} = \frac{1}{2}(\varepsilon_{[110]} - \varepsilon_{[\bar{1}10]})$, where we use the simplified notation $\varepsilon_x \equiv \varepsilon_{xx}$. Similar expressions apply relating $R_\alpha$ with $R_{xx}$. The symmetry-decomposed elastoresistance coefficients are then defined as $m_\alpha = (1/R_\alpha)dR_\alpha/d\varepsilon_\alpha$. In terms of the experimental data, these coefficients are given by

$$m_{A_{1g}} = \frac{1}{1-\nu_{[100]}}\left[\frac{1}{R_{[100]}}\frac{dR_{[100]}}{d\varepsilon_{[100]}} + \frac{1}{R_{[010]}}\frac{dR_{[010]}}{d\varepsilon_{[100]}}\right] \quad (1a)$$

$$= \frac{1}{1-\nu_{[110]}}\left[\frac{1}{R_{[110]}}\frac{dR_{[110]}}{d\varepsilon_{[110]}} + \frac{1}{R_{[\bar{1}10]}}\frac{dR_{[\bar{1}10]}}{d\varepsilon_{[110]}}\right] \quad (1b)$$

| | $c_{11}$ | $c_{12}$ | $c_{13}$ | $c_{33}$ | $c_{66}$ | $\nu_{[100]}$ | $\nu'_{[100]}$ | $\nu_{[110]}$ | $\nu'_{[110]}$ |
|---|---|---|---|---|---|---|---|---|---|
| $Ba_{0.5}K_{0.5}Fe_2As_2$ | 123.1 | 51.3 | 66.2 | 92.2 | 51.0 | 0.049 | 0.682 | $-0.126$ | 0.808 |
| $KFe_2As_2$ | 79.7 | 46.6 | 38.4 | 45.1 | 31.5 | 0.296 | 0.600 | $-0.017$ | 0.866 |
| $RbFe_2As_2$ | 78.3 | 40.2 | 28.4 | 53.8 | 33.2 | 0.398 | 0.318 | 0.143 | 0.453 |
| $CsFe_2As_2$ | 84.4 | 45.9 | 39.5 | 61.2 | 31.6 | 0.346 | 0.422 | 0.113 | 0.572 |

**Table 1 Table of elastic constants (in units of GPa) and Poisson ratios (unitless), obtained from ab initio calculations.**

$$m_{B_{1g}} = \frac{1}{1+\nu_{[100]}}\left[\frac{1}{R_{[100]}}\frac{dR_{[100]}}{d\varepsilon_{[100]}} - \frac{1}{R_{[010]}}\frac{dR_{[010]}}{d\varepsilon_{[100]}}\right] \quad (1c)$$

$$m_{B_{2g}} = \frac{1}{1+\nu_{[110]}}\left[\frac{1}{R_{[110]}}\frac{dR_{[110]}}{d\varepsilon_{[110]}} - \frac{1}{R_{[\bar{1}10]}}\frac{dR_{[\bar{1}10]}}{d\varepsilon_{[110]}}\right]. \quad (1d)$$

The quantity $\nu$ is the sample's Poisson ratio, which relates the strain along different directions, $\nu_{[100]} = -\varepsilon_{[010]}/\varepsilon_{[100]}$, and $\nu_{[110]} = -\varepsilon_{[\bar{1}10]}/\varepsilon_{[110]}$. In general, $\nu$ is sample and temperature-dependent and anisotropic ($\nu_{[100]} \neq \nu_{[110]}$). The elastoresistance coefficients in the notation of irreducible representations are given in terms of the elastoresistance coefficients in the usual Voigt notation as

$$m_{B_{1g}} = m_{11} - m_{12} \quad (2a)$$

$$m_{B_{2g}} = 2m_{66} \quad (2b)$$

$$m_{A_{1g}} = (m_{11} + m_{12}) - \frac{2\nu'}{1-\nu}m_{13} \quad (2c)$$

$$= m_{A_{1g,1}} - \frac{2\nu'}{1-\nu}m_{A_{1g,2}} \quad (2d)$$

In the $A_{1g}$ channel, $m_{13} = m_{A_{1g,2}}$ contributes and cannot be disentangled from $m_{11} + m_{12} = m_{A_{1g,1}}$. Our measured elastoresistance coefficient $m_{A_{1g}}$ therefore includes the effect of $c$-axis compression (tension) which accompanies the symmetric in-plane biaxial tension (compression) because of the Poisson effect.

We note that the reported elastoresistance coefficients for $CsFe_2As_2$ are larger in this paper than in our previous work[45]. This results from a difference in the ratio between real and nominal strain in the two studies. Our previous work was conducted with a Razorbill CS-100 cell, which has a smaller maximum displacement than the CS-120 cell used here. According to the documentation provided with our strain cells by Razorbill Instruments, due to the deformation of the strain device itself by a stiff sample, the capacitance sensor will overread the applied displacement according to

$$\frac{\Delta L_{\mathrm{meas}}}{\Delta L_{\mathrm{real}}} = 1 + \frac{k_s}{k_\tau}(h_s + h_a)(h_s + h_c), \quad (3)$$

where $k_s$ is the sample spring constant, and $h_s$ is the sample mounting height above the top of the apparatus. The remaining parameters are properties of the strain cell itself: $k_\tau$ is the torsional spring constant, $h_a$ is the height of the top surface above the moving block rotation center, and $h_c$ is the height of the top surface above the center of the capacitor. The parameters $k_\tau$, $h_a$ and $h_c$ are different between the two strain cells. In addition, the sample spring constants $k_s$ in this work were typically ~2 N/μm, compared with ~4 N/μm in our previous work. The smaller $k_s$ allows for a transmitted strain closer to the nominal value determined from the capacitance sensor, producing a greater measured elastoresistance in our current work. Due to these issues, we chose to re-measure $CsFe_2As_2$ in the CS-120 cell to make sure the results were comparable in all presented samples.

**Calculation of Poisson ratios**. The sample Poisson ratios $\nu_x = -\epsilon_{yy}/\epsilon_{xx}$ and $\nu'_x = -\epsilon_{zz}/\epsilon_{xx}$ in terms of the elastic constants $c_{ij}$ are given by

$$\nu_{[100]} = \frac{c_{13}^2 - c_{12}c_{33}}{c_{13}^2 - c_{11}c_{33}} \quad (4a)$$

$$\nu'_{[100]} = \frac{(c_{12} - c_{11})c_{13}}{c_{13}^2 - c_{11}c_{33}} \quad (4b)$$

$$\nu_{[110]} = \frac{c_{33}(c_{11} + c_{12} - 2c_{66}) - 2c_{13}^2}{c_{33}(c_{11} + c_{12} + 2c_{66}) - 2c_{13}^2} \quad (4c)$$

$$\nu'_{[110]} = \frac{4c_{13}c_{66}}{c_{33}(c_{11} + c_{12} + 2c_{66}) - 2c_{13}^2}. \quad (4d)$$

To obtain rough estimates of the elastic constants, we performed ab initio calculations in the framework of the generalized gradient approximation using a mixed-basis pseudopotential method[36]. The phonon dispersions and corresponding interatomic force constants were calculated via density functional perturbation theory and the elastic constants were then extracted via the method of long waves. For $Ba_{0.4}K_{0.6}Fe_2As_2$, we used values calculated for $Ba_{0.5}K_{0.5}Fe_2As_2$ in a virtual-crystal approximation. The numerical values of elastic constants and Poisson ratios calculated therefrom are given in Table 1. For the purposes of calculating the elastoresistance coefficients, the Poisson ratios were taken to be temperature independent.

## Data availability
The data that support the findings of this study are available from the corresponding author upon reasonable request.

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

## Acknowledgements
We thank I.R. Fisher and K. Grube for valuable discussions. We acknowledge support by the Helmholtz Association under Contract No. VH-NG-1242. This work was also supported by the German Research Foundation (DFG) under CRC/TRR 288 (Project A02).

## Author contributions
P.W. and A.E.B. initiated the project. A.A.H. and T.W. grew single-crystal samples. P.W., A.E.B., and M.F. performed the experiments. R.H. performed DFT calculations of elastic constants. P.W. analyzed the data and P.W., C.M., and A.E.B. developed the interpretations. A.E.B. guided the project. P.W. and A.E.B. wrote the manuscript with input from all authors.

## Funding

## Competing interests
The authors declare no competing interests.
