## [Peer Review File · Nature Communications]

REVIEWER COMMENTS

Reviewer #1 (Remarks to the Author):

The current manuscript presents a symmetry resolved elastoresistance measurement on the heavily hole doping 122 iron pnictides, including 60% K-doped-Ba122, KFe_2As_2 , RbFe_2As_2 and CsFe_2As_2 . The authors applied uni-axial stress along the 110 and 100 direction and measure the strain dependence of the resistance along the longitudinal and transverse direction. Through a careful analysis, they discovered that the elastoresistance is dominated by the symmetric A1g response as approaching towards CsFe_2As_2 . There is a remanent B2g response in K-doped-Ba122 but it is progressively weakened in KFe_2As_2 , RbFe_2As_2 and CsFe_2As_2 . In addition, very weak B1g response was observed with no evidence of B1g nematic phase transition. Finally, by analyzing the temperature dependence of resistivity under different strain, the author concluded that the A1g elastoresistance is equivalent to the strain derivative of the Sommerfeld coefficient, or the Grüneisen parameter. This paper presents a high-quality study on an important topic of iron-based superconductors. It expands on the author's previous work on CsFe_2As_2 and further clarify the issue of B1g nematicity in the heavily hole doped 122 pnictides. It also demonstrated a new capability of elastoresistivity – the measurement of the strain derivative of Sommerfeld coefficient. This new capability will be particularly useful for systems close to quantum criticality. Therefore, I recommend the publication of this paper. Below are two questions/comments that might help improve this paper:

1. The zero-strain Sommerfeld coefficient of KFe_2As_2 , RbFe_2As_2 , CsFe_2As_2 is well characterized, for example it was reported in the ref 36 of the current manuscript. In principle one could construct the function of how Sommerfeld coefficient depends on in-plane lattice constant by interpolating these three data points. From this function one could derive the strain derivative of Sommerfeld coefficient and compare with that measured by the A1g elastoresistance. The author may consider performing this analysis and see if the numbers are consistent.
2. The C_{66} and Poisson ratio of K-doped-Ba122 should be strongly temperature dependent due the B2g nematic fluctuations. However, reading the method section it looks like the authors assumed a temperature independent c_{66} and Poisson ratio. How does this assumption influence the accuracy of extracted B2g elastoresistance?

Reviewer #2 (Remarks to the Author):

The paper by Wiechi et. al. reports on their strain depend resistance measurements of several hole-doped iron-based superconductors, namely the end-members AFe_2As_2 ($\text{A} = \text{K}, \text{Rb}, \text{Cs}$) and one over-doped $\text{K}_{1-x}\text{Ba}_x\text{Fe}_2\text{As}_2$ at $x=0.6$. The authors analyzed their data by decomposing the strain response into different symmetry channels, and found the symmetric A1g channel dominates in AFe_2As_2 , while B2g dominates in $\text{K}_{0.6}\text{Ba}_{0.4}\text{Fe}_2\text{As}_2$. The nematicity in the B1g channel was found very small in all investigated samples. The authors relates their results to the presumable critical point in AFe_2As_2 , to the strong electronic correction in hole-doped iron-based compounds, and to the effect of nematic fluctuation on the superconducting temperature T_c . The manuscript is well written, and the figures are well prepared.

In order to evaluate the importance of this paper, we should be aware that the main conclusions of this paper are not new. There are already quite a few publications on this topic. The same team used the same technique and reported the strain dependent resistance of CsFe_2As_2 is dominated by the A1g symmetry channel in Ref. 45. In my opinion, this paper is a following research of Ref. 45. On the other hand, the conclusions of this paper are in sharp contrast to another paper (Ref. 39) which also reported the strain dependent resistance of AFe_2As_2 . The authors of Ref. 39 reported

huge B1g nematicity in AFe_2As_2 , and a XY-nematicity was proposed for the hole doped $\text{Rb}_{1-x}\text{Ba}_x\text{Fe}_2\text{As}_2$ (effectively the same as $\text{K}_{1-x}\text{Ba}_x\text{Fe}_2\text{As}_2$ involved in this paper). Since the two conflicting papers used very similar experimental technique, at least one of them is wrong at the raw data level.

Wiechi et. al. seem to be quite confident of their results. In Ref. 45, they pointed out that large difference between the strain generators and the AFe_2As_2 crystals cause tiny cracks in the samples. They concluded it is of crucial importance that the tension created by the difference between samples and strain generators should be carefully compensated. However, I do not think they have convinced, at least the fans of Ref. 39, that the (only) correct way to perform strain dependent resistance measurement is as what they did. It is anticipated this manuscript should provide more indisputable proof that what they report, instead of Ref. 39, is correct. As the authors have written in the abstract, it will revise the current understanding of this topic. In my opinion, only under this condition, this paper is worthwhile to appear in Nature Communication. Otherwise, it just contributes to the ongoing fight between a handful of groups in this field, and will not attract much attention from the outside.

Please see below the arguments and suggestions toward this direction:

1. The main fact that endow the authors to depreciate the results of Ref. 39 is that samples might get tiny cracks during the cooling down process. According to the authors' previous paper (Ref. 45), such crack happens at ~ 200 K for CsFe_2As_2 . According to Fig. 1b of this paper, that corresponds to a tension strain of $\sim 0.15\%$. One may notice that a strain up to 0.24% was applied to KFe_2As_2 in this study (Fig.3). Does it mean KFe_2As_2 is more rigid than CsFe_2As_2 ?

Even if it is true that CsFe_2As_2 and RbFe_2As_2 crack due to uncompensated tension strain during the cooling process, elastoresistance can still be properly measured. That is because elastoresistance is measured at fixed temperatures. As long as the small strain applied to the (cracked) crystal does not further break the sample or heal the flaw, elastoresistance is not affected. The good linear relationship between resistance and strain presented in Ref. 39 proven their samples are under such condition. I would like to suggest Wiechi et. al. try whether the results of Ref. 39 can be reproduced based on the same technique? According to arXiv 2011.13207, the authors are capable to do such experiments. They can also try to use their current set up but not compensate the tension strain, and see whether the resistance still grows linearly with strain. If so, will the calculated elastoresistance become very different?

2. As indicted in Fig. 1d, the authors measured longitudinal and transverse elastoresistance on the same sample. Note that they measured on two separated bar shaped samples in Ref. 45. While I think the longitudinal measurement is fine, the transverse measurement could have some problems. With stripes of conductive paints shortcut two edges, can the 'transverse' contacts pick up the real transverse resistance? I appreciate the authors for reporting their new set of data instead of taking their previous data of CsFe_2As_2 from Ref.45, that makes a comparison possible. The A1g nematicity of CsFe_2As_2 reported in this new paper (Fig.5a) is 60% larger than that in Ref. 45 (Fig.2e), and the B2g nematicity looks totally different. That raises the concern to what extent should we trust such results. Does the different way how contacts were made cause such mismatching? According to Ref. 1, the modified Montgomery contact is ideal for elastoresistance measurements. I would like to suggest the authors to try this modified Montgomery configuration on at least one selected sample to verify their present results. Note that the Montgomery configuration was tested on one sample in Ref. 39, and their conclusion is that the A1g contribution is negligibly small.

3. The authors stated that they use DevCon 5 minutes Epoxy to mount their samples on the strain cell. The DevCon Epoxy is very soft. The effective strain transmitted to the sample was verified for gluing the sample on the surface of a piezo stack in Ref. 25. For Razorbill-type strain cells, a much stronger epoxy Stycast is frequently used. Stycast has been verified to be a good choice to transmit full strain to the sample (see, for example Nat Mater 16 708). As far as I know, such test has not been done for DevCon Epoxy with Razorbill cells. That makes one wonder whether the samples experience the strain applied to it. For example in Ref. 2a, and 2c, some of the black data points are surprisingly close to

zero, which is strange. The deformation of a simple metal plate will result in an elastoresistance of the order of 1. A measured elastoresistance near 0 may imply the sample does not feel strain at all. I recommend the authors either demonstrate DevCon works as well as Stycast or simply try to use Stycast for at least one of their samples.

Besides, some issues should be clarified before the paper is published

1. In the paragraph from line #69 to line #75, the authors refuted the attempt to associate a phase transition with the drops in the elastoresistance in Ref. 39. Based on the fact that no hint of phase transition is seen in $R(T)$ or $L(T)$ curves. In my opinion, it is a strong argument against a phase transition. However, I think the interpretation for such drop should be discussed in more details. Note that the elastoresistance of many samples with much lower RRR than the AFe_2As_2 samples have been reported in the literatures, but a drop in the elastoresistance is very rare. For example, the resistance of $\text{Ba}(\text{Fe}_{0.955}\text{Ni}_{0.045})_2\text{As}_2$ reported in Ref. 1 becomes comparable to its residual resistance at ~ 50 K (Fig. S11), but only a slight drop in elastoresistance is presented just above the T_c (~ 20 K) of this sample (Fig. 2B). Wiechi et. al. also mentioned the drop of elastoresistance can be a result of the coherence-incoherence crossover (They emphasized such idea in the paragraph from line #145 to line #149). However, in my opinion, it is not straightforward to accept this claim just based on their similar temperature range. The authors should explain how these two phenomena are related.

2. I do not think it is reasonable to treat AFe_2As_2 and $\text{K}_{1-x}\text{Ba}_x\text{Fe}_2\text{As}_2$ as a continues doping series (Fig. 6 and the related discussion parts in the main text). That is because (i) Substituting K to Ba is hole doping, while AFe_2As_2 is isovalent doping (ii) The lattice parameter a decreases from BaFe_2As_2 to KFe_2As_2 , while it increases from KFe_2As_2 to CsFe_2As_2 . If the authors wish to show some decent doping dependence, I think they should measure more samples of $\text{K}_{1-x}\text{Ba}_x\text{Fe}_2\text{As}_2$ with different x , and some samples like $\text{K}_{1-x}\text{Cs}_x\text{Fe}_2\text{As}_2$.

3. In vein of my previous comment, the 'monotonic' increase of A_{1g} response and the 'weakening' of B_{2g} response with hole doping as the authors stated in the introduction part (line #44) is only based on two points, $x=0.6$ and $x=1.0$. I do not think any trend can be summarized based on two points. Furthermore, if one includes the $x=0.4$ case reported in Ref. 1, he will find that the B_{2g} response is nearly the same as $x=0.6$ reported in this paper. So, there is no 'monotonic' trend. This raise another question, why B_{2g} is not reduced from $x=0.4$ to $x=0.6$? It is accepted that nematic fluctuation is strong at the critical point ($x=0.4$), and should decrease when it is away from the criticality. However, it seems not to be the case in $\text{K}_{1-x}\text{Ba}_x\text{Fe}_2\text{As}_2$.

4. Additionally, it is far-fetched to claim that the extended superconducting dome in $\text{K}_{1-x}\text{Ba}_x\text{Fe}_2\text{As}_2$ is related to the remnants of the B_{2g} nematicity, as the authors wrote from line #166. Nematicity is expected to enhance superconductivity only for certain types of pairing. The superconducting states of hole-doped $\text{K}_{1-x}\text{Ba}_x\text{Fe}_2\text{As}_2$ is very complicated. It is likely change from S_{+-} to s_{+is} to nodal (probably d -wave) with increasing x . There is even a Lifshitz transition around $x=0.8$, which the authors also pointed out. It is hard to believe the B_{2g} nematicity can be a determinative factor to T_c in such a complicated situation. It is even no consensus whether nematicity matters at all in any of these superconducting states.

5. In the paragraph from line #84 to line #90, the authors pointed out that the tensile strain has the same effect as the negative chemical pressure (from $A = \text{K}$ to Cs) on the parameter A . However, as they presented, T_c of KFe_2As_2 increases under tensile strain, while decreasing with negative chemical pressure. In that sense, is the same trend presented by parameter A just a coincidence?

6. The authors mentioned in line #199 that DuPont 4929N sliver paint were found to be mechanically unstable on strain application. I think it would be helpful if the authors can provide more details in the supplementary materials. As far as I know, DuPont 4929N is wildly used in transport measurements, including many of the previous elastoresistance studies. Does it mean all the results of the previous publication involved contacts made with DuPont 4929N need to be revisited?

7. In the paragraphs from line #91 to line #106, the authors redefined the elastoresistance m and

argued it produces more physically meaningful results. I think it is an interesting finding. The origin of the elastoresistance can be anisotropic scattering rate or Pomeranchuk instability of the Fermi surface. As far as I know, there is no conclusion on the leading factor of nematicity in iron-based superconductors yet. The analysis of this paper seems to be a strong support of the scattering scenario. Could the authors discuss in more detail on this point? Again, does it mean we should re-analysis all the published elastoresistance results to get this redefined m , in order to get more physically meaningful results? Note that the RRR values of other iron-based superconductors are much smaller than AFe_2As_2 . So, the calculated elastoresistance of other iron-based superconductors will change drastically if R_0 is considered.

To conclude, I do not recommend this paper to appear on Nature Communication in its current form.

Reviewer #1 (Remarks to the Author):

The current manuscript presents a symmetry resolved elastoresistance measurement on the heavily hole doping 122 iron pnictides, including 60% K-doped-Ba122, KFe2As2, RbFe2As2 and CsFe2As2. The authors applied uni-axial stress along the 110 and 100 direction and measure the strain dependence of the resistance along the longitudinal and transverse direction. Through a careful analysis, they discovered that the elastoresistance is dominated by the symmetric A1g response as approaching towards CsFeAs2. There is a remanent B2g response in K-doped-Ba122 but it is progressively weakened in KFe2As2, RbFe2As2 and CsFe2As2. In addition, very weak B1g response was observed with no evidence of B1g nematic phase transition. Finally, by analyzing the temperature dependence of resistivity under different strain, the author concluded that the A1g elastoresistance is equivalent to the strain derivative of the Sommerfeld coefficient, or the Gruneisen parameter.

This paper presents a high-quality study on an important topic of iron-based superconductors. It expands on the author's previous work on CsFe2As2 and further clarify the issue of B1g nematicity in the heavily hole doped 122 pnictides. It also demonstrated a new capability of elastoresistivity – the measurement of the strain derivative of Sommerfeld coefficient. This new capability will be particularly useful for systems close to quantum criticality. Therefore, I recommend the publication of this paper.

We thank the referee for their recommendation to publish our paper in Nature Communications.

Below are two questions/comments that might help improve this paper:

1. The zero-strain Sommerfeld coefficient of KFe2As2, RbFe2As2, CsFe2As2 is well characterized, for example it was reported in the ref 36 of the current manuscript. In principle one could construct the function of how Sommerfeld coefficient depends on in-plane lattice constant by interpolating these three data points. From this function one could derive the strain derivative of Sommerfeld coefficient and compare with that measured by the A1g elastoresistance. The author may consider performing this analysis and see if the numbers are consistent.

As pointed out by the referee, one can calculate the strain derivative of the Sommerfeld coefficient γ in the AFe₂As₂ series from data under ambient conditions. We need only the in-plane lattice parameter and γ . For A=K we have (3.8441 Å, 101.8 mJ/molK²). Similarly, for A=Rb (3.872 Å, 127.0 mJ/molK²) and A=Cs (3.9017 Å, 180.2 mJ/molK²). From these data, one can calculate $\frac{\Delta\gamma}{\Delta\varepsilon} = 3.471$ J/molK² between K and Rb. Similarly, between Rb and Cs one can calculate $\frac{\Delta\gamma}{\Delta\varepsilon} = 6.891$ J/molK². Another way to estimate $d\gamma/d\varepsilon$ is to extract it directly from thermal expansion measurements. In this way, Eilers *et al.* [Phys. Rev. Lett. **116** 237003 (2016)], find that $\frac{d\gamma}{d\varepsilon} = 1.15$ J/molK² for A=K, $\frac{d\gamma}{d\varepsilon} = 2.49$ J/molK² for A=Rb and $\frac{d\gamma}{d\varepsilon} = 7.74$ J/molK² for A=Cs. In the section "Quantitative

comparison between elastoresistance and thermodynamic quantities” of our Supplemental Material, we show that these values are in excellent agreement with the elastoresistance data, using a Kadowaki-Woods relation appropriate for this system.

2. The C_{66} and Poisson ratio of K-doped-Ba122 should be strongly temperature dependent due the B2g nematic fluctuations. However, reading the method section it looks like the authors assumed a temperature independent c_{66} and Poisson ratio. How does this assumption influence the accuracy of extracted B2g elastoresistance?

In the case of BaFe_2As_2 , the elastic constant c_{66} shows a large softening of at least 80% - 90% above the magnetic/nematic transition at 135 K, leading to a strong temperature dependence of the Poisson ratio $\nu_{[110]}$. As we now show in Supplementary Fig. S1, this temperature dependence of the Poisson ratio must be considered to extract the B_{2g} elastoresistance coefficient in BaFe_2As_2 . However, in the heavily-hole-doped $\text{K}_x\text{Ba}_{1-x}\text{Fe}_2\text{As}_2$ samples ($x > 0.6$), the softening of c_{66} is much weaker ($\sim 10\%$ or less) [see, for example, Fig. 1d of A. E. Böhrer *et al.*, Phys. Rev. Lett. **112**, 047001 (2014)]. Therefore, in the heavily-hole-doped compounds, a temperature independent $\nu_{[110]}$ is a reasonable approximation. To our knowledge, no data exist on these samples which would allow a computation of the temperature dependence of $\nu_{[110]}$.

Reviewer #2 (Remarks to the Author):

The paper by Wiecki et. al. reports on their strain depend resistance measurements of several hole-doped iron-based superconductors, namely the end-members AFe_2As_2 ($A = \text{K}, \text{Rb}, \text{Cs}$) and one over-doped $\text{K}_{1-x}\text{Ba}_x\text{Fe}_2\text{As}_2$ at $x=0.6$. The authors analyzed their data by decomposing the strain response into different symmetry channels, and found the symmetric A_{1g} channel dominates in AFe_2As_2 , while B_{2g} dominates in $\text{K}_{0.6}\text{Ba}_{0.4}\text{Fe}_2\text{As}_2$. The nematicity in the B_{1g} channel was found very small in all investigated samples. The authors relates their results to the presumable critical point in AFe_2As_2 , to the strong electronic correction in hole-doped iron-based compounds, and to the effect of nematic fluctuation on the superconducting temperature T_c . The manuscript is well written, and the figures are well prepared.

In order to evaluate the importance of this paper, we should be aware that the main conclusions of this paper are not new. There are already quite a few publications on this topic. The same team used the same technique and reported the strain dependent resistance of CsFe_2As_2 is dominated by the A_{1g} symmetry channel in Ref. 45. In my opinion, this paper is a following research of Ref. 45.

We would like to thank the referee for their careful reading and detailed comments. While it is true that we have published similar data on the single material CsFe_2As_2 elsewhere [P. Wiecki *et al.*, Phys. Rev. Lett. **125**, 187001 (2020), "Ref. 45"], we do not try to hide this from the readers or editors of Nature Communications. In this previous work, we reported on the dominant A_{1g} elasto-resistance of CsFe_2As_2 . However, this work was insufficient to resolve the detailed temperature dependence of the B_{1g} and B_{2g} elasto-resistance coefficients and did not track their evolution with doping and chemical substitution. In consequence, while our initial data on CsFe_2As_2 were inconsistent with Ishida *et al.* [Proc. Nat. Acad. Sci. **117** 6424 (2020), "Ref. 39"], we can only now address the claims of Ishida *et al.* regarding RbFe_2As_2 and XY-nematicity in the hole-doping series. We therefore believe that this "follow-up" has a high-impact on its own.

On the other hand, the conclusions of this paper are in sharp contrast to another paper (Ref. 39) which also reported the strain dependent resistance of AFe_2As_2 . The authors of Ref. 39 reported huge B_{1g} nematicity in AFe_2As_2 , and a XY-nematicity was proposed for the hole doped $\text{Rb}_{1-x}\text{Ba}_x\text{Fe}_2\text{As}_2$ (effectively the same as $\text{K}_{1-x}\text{Ba}_x\text{Fe}_2\text{As}_2$ involved in this paper). Since the two conflicting papers used very similar experimental technique, at least one of them is wrong at the raw data level.

We appreciate the referee's concern with the conflicting statements. Concerning the disagreement with the previous study, we believe that the raw data is not the main discrepancy between the two works, although a few notable differences in the raw data do exist. Instead, we need to respectfully point out that the work of Ishida *et al.* has a clear conceptual flaw. Namely, only the longitudinal elasto-resistance was measured,

and not the transverse, except for a single sample of $\text{Ba}_{0.35}\text{Rb}_{0.65}\text{Fe}_2\text{As}_2$. Furthermore, the longitudinal elastoresistance was simply referred to as nematic susceptibility χ_{nem} . This is incorrect in general. As detailed in the Methods (section “Elastoresistance measurements”), the longitudinal elastoresistance for strain applied along [100] is given by:

$$\frac{1}{R_{[100]}} \frac{dR_{[100]}}{d\varepsilon_{[100]}} = \left(\frac{1 - \nu_{[100]}}{2} \right) m_{A_{1g}} + \left(\frac{1 + \nu_{[100]}}{2} \right) m_{B_{1g}}$$

Similarly, when strain is applied along [110], the longitudinal elastoresistance is

$$\frac{1}{R_{[110]}} \frac{dR_{[110]}}{d\varepsilon_{[110]}} = \left(\frac{1 - \nu_{[110]}}{2} \right) m_{A_{1g}} + \left(\frac{1 + \nu_{[110]}}{2} \right) m_{B_{2g}}$$

Thus, the longitudinal elastoresistance generally measures a linear combination of the A_{1g} and B_{1g}/B_{2g} elastoresistance coefficients. The B_{1g} (B_{2g}) nematic susceptibility is proportional to $m_{B_{1g}}$ ($m_{B_{2g}}$) [H. Kuo *et al.*, Science **352**, 958 (2016)]. In order to measure nematic susceptibility, one needs to carefully isolate $m_{B_{1g}}$ or $m_{B_{2g}}$ by measuring both the transverse and longitudinal elastoresistance. However, Ishida *et al.* have not attempted to isolate $m_{B_{1g}}$ or $m_{B_{2g}}$. Instead, Ishida *et al.* simply assumed $m_{A_{1g}} = 0$ which is a good approximation only for some compounds, and therefore, the large in-plane symmetric response was incorrectly identified as nematic. This appears to be ultimately the primary reason for our difference of interpretation. Since this fact appears to have been overlooked by the referee, we have added an explicit statement in the manuscript.

Since Ishida *et al.* did not measure the transverse elastoresistance, we can only compare the raw longitudinal elastoresistance data. As discussed below, the true strain in our strain cell is approximately 60% of the nominal strain (that is, $\varepsilon_{true} = 0.6 \varepsilon_{nom}$). In contrast, for measurements on a piezo stack, like those of Ishida *et al.*, the strain is presumably fully transmitted [H. Kuo *et al.*, Science **352** 958 (2016)]. To correct for this, we scale up our elastoresistance data by a factor of (1/0.6), as in Supplementary Fig. S1 of our revised manuscript. The resulting direct comparison of longitudinal elastoresistance data is shown in Fig. R1 of this response letter.

As seen in Fig. R1, in many cases our raw longitudinal elastoresistance data are in good agreement with those of Ishida *et al.*. For example, when we compare our data for $\text{K}_{0.6}\text{Ba}_{0.4}\text{Fe}_2\text{As}_2$ with the data of Ishida *et al.* for $\text{Rb}_{0.65}\text{Ba}_{0.35}\text{Fe}_2\text{As}_2$ (similar doping level) for strain applied along [110] (blue curves in Fig. R1a), we find good agreement. Similarly, we compare our data for RbFe_2As_2 with the data of Ishida *et al.* for both RbFe_2As_2 and $\text{Rb}_{0.86}\text{Ba}_{0.14}\text{Fe}_2\text{As}_2$. Again, reasonable agreement is found for strain applied along [110] (blue curves in Fig. R1b,c). Thus, at the raw data level, the two data sets are not as different as they may appear at first glance.

The agreement is not as good when strain is applied along [100], however. In particular, the data of Ishida *et al.* for pure RbFe₂As₂ (Fig. R1c, open red triangles) are vastly larger than ours. This particular dataset is also much larger than Ishida *et al.*'s own data for the 86% Rb-doped sample, as highlighted in Fig. R2. We note that such a large jump of the longitudinal elastoresponse between 86% and 100% Rb doping may be implausible.

Fig. R1 Comparison of our longitudinal elastoresponse data with those of Ishida *et al.* [Proc. Nat. Acad. Sci. **117** 6424 (2020)]. Our elastoresponse data is shown in filled symbols, while data from Ishida *et al.* is shown in open symbols. Our data have been scaled upwards by a factor of $(1/0.6)$ to account for the fact that $\epsilon_{true} = 0.6 \epsilon_{nom}$ in our strain cell.

We appreciate that the referee is screening our data for plausibility and reliability. However, we ask that the referee apply the same scrutiny to the data of Ishida *et al.* also when considering the remaining differences in the raw data sets. The soundness of our raw data is supported by the consistency of our conclusions with well-established thermodynamic measurements on these materials, as we have detailed in our Supplementary Information (see the sections "Quantitative comparison between elastoresponse and thermodynamic quantities" and "Qualitative comparison between elastoresponse and thermal expansion").

Fig. R2 Comparison of Ishida *et al.* data for RbFe_2As_2 and $\text{Rb}_{0.86}\text{Ba}_{0.14}\text{Fe}_2\text{As}_2$.

Wiechi *et al.* seem to be quite confident of their results. In Ref. 45, they pointed out that large difference between the strain generators and the AFe_2As_2 crystals cause tiny cracks in the samples. They concluded it is of crucial importance that the tension created by the difference between samples and strain generators should be carefully compensated. However, I do not think they have convinced, at least the fans of Ref. 39, that the (only) correct way to perform strain dependent resistance measurement is as what they did. It is anticipated this manuscript should provide more indisputable proof that what they report, instead of Ref. 39, is correct. As the authors have written in the abstract, it will revise the current understanding of this topic. In my opinion, only under this condition, this paper is worthwhile to appear in Nature Communications. Otherwise, it just contributes to the ongoing fight between a handful of groups in this field, and will not attract much attention from the outside. Please see below the arguments and suggestions toward this direction:

We would like to thank the reviewer agreeing with us that our data, if correct, will revise the current understanding of this topic and will be worthy of publication in Nature Communications in this case.

Below, we explain why we are confident about our data. However, we would like to point out that we do not claim that our strain-cell-based method is the only correct way to perform elastoresistance measurements. In fact, for samples with low thermal expansion, such as BaFe_2As_2 , our technique produces the same results as traditional piezo-stack-based elastoresistance measurements, up to a factor resulting from different strain transmission. To give the referee additional confidence in our method, we have now included elastoresistance data of BaFe_2As_2 measured with the strain-cell approach in our Supplemental Information (Supplementary Fig. S1 of the revised manuscript). As explained in the manuscript and illustrated in Fig. 1b of the manuscript, the fully hole-doped samples have a particularly extreme thermal expansion, necessitating the use of a strain cell for elastoresistance measurements. (The extreme

thermal expansion is, in fact, a direct consequence of the highly strain sensitive electronic entropy which we measure in this work.)

1. The main fact that endow the authors to depreciate the results of Ref. 39 is that samples might get tiny cracks during the cooling down process. According to the authors' previous paper (Ref. 45), such crack happens at ~ 200 K for CsFe₂As₂. According to Fig. 1b of this paper, that corresponds to a tension strain of $\sim 0.15\%$. One may notice that a strain up to 0.24% was applied to KFe₂As₂ in this study (Fig.3). Does it means KFe₂As₂ is more rigid than CsFe₂As₂?

As discussed in detail above, the primary reason we discount the conclusions of Ishida *et al.* is that the lack of transverse elastoresistance data means that they have not actually measured the m_{B1g} and m_{B2g} elastoresistance coefficients, which are required to make any conclusions about nematicity. Indeed, in our previous work ["Ref. 45"] we investigated what happens to CsFe₂As₂ samples when they are glued to a substrate of low thermal expansion and found that samples likely sustain small cracks on cooling. We believe there are two reasons why we can apply a larger strain here and we do not believe that this is because KFe₂As₂ is more "rigid" than the CsFe₂As₂. It is our experience that samples can withstand greater tension strains at low temperatures. In this work, large strains ($\sim 0.2\%$) were applied only at low temperatures (below ~ 20 K), whereas samples glued to glass substrates in our previous work experience such a strain at 200 K. Another reason why we can apparently achieve higher strains in the strain cell than we can for samples glued on the piezo may have to do with the different character of the strains in the two cases. A sample glued to a piezoelectric stack or glass plate experiences a biaxial tension stress due to the thermal expansion mismatch, leading to a biaxial tension strain. In contrast, samples in the strain cell experience a strictly uniaxial tension stress. The sample responds by freely contracting along the perpendicular direction due to the Poisson effect, leading to an anisotropic strain. The sample therefore experiences very different strain conditions in the two setups, and may therefore yield at different nominal strains.

Even if it is true that CsFe₂As₂ and RbFe₂As₂ crack due to uncompensated tension strain during the cooling process, elastoresistance can still be properly measured. That is because elastoresistance is measured at fixed temperatures. As long as the small strain applied to the (cracked) crystal does not further break the sample or heal the flaw, elastoresistance is not affected. The good linear relationship between resistance and strain presented in Ref. 39 proven their samples are under such condition. I would like to suggest Wiechi et. al. try whether the results of Ref. 39 can be reproduced based on the same technique? According to arXiv 2011.13207, the authors are capable to do such experiments. They can also try to use their current set up but not compensate the tension strain, and see whether the resistance still grows linearly with strain. If so, will the calculated elastoresistance become very different?

We do not believe that there is clear proof for the referee's assertion that the intrinsic elastoresistance can be properly measured on a damaged sample. We believe it is more plausible that a damaged sample no longer responds purely elastically to strain. Once the sample cracks, the actual current path can equally well be modulated by external stress in addition to any intrinsic elastoresistance. Furthermore, a linear resistance vs. strain relationship does not necessarily prove the sample is undamaged, particularly for the very small strain sweeps $\Delta\varepsilon < 0.02\%$ obtainable by the piezoelectric-stack method at low temperatures.

The referee suggests that we attempt to reproduce our results using the piezoelectric-stack method rather than the strain-cell method. In fact, we have already made efforts in this direction in our previous work on CsFe_2As_2 [see Supplementary Material of P. Wiecki *et al.*, Phys. Rev. Lett. **125**, 187001 (2020)]. In these initial attempts, we found data qualitatively similar to the data of Ishida *et al.*, but with clear signs of sample damage. We are also aware of unpublished data on CsFe_2As_2 acquired via the piezoelectric-stack method which differ from both our measurement and Ishida *et al.*'s measurement [X.C. Hong and C. Hess (IFW Dresden), private communication], suggesting difficulties with reproducibility using this technique on these samples. In addition to the thermal expansion mismatch with the piezoelectric stack, these samples are also mechanically very soft and easily bent compared to BaFe_2As_2 . The samples may therefore be subject to bending and damage during the sample mounting, which requires pressing the entire sample down into the epoxy several times as the epoxy cures.

Nevertheless, we tried again to reproduce our strain-cell data by the piezoelectric-stack method. As detailed above, in many cases our strain-cell data already agree with those of Ishida *et al.*, therefore we focus here on one particular case where the disagreement is large, namely the [100] longitudinal for RbFe_2As_2 . Our data are shown in Fig. R3. As before, the strain-cell data have been multiplied by (1/0.6) in order to correct for the non-full strain transmission in the strain cell (supplementary Fig. S1 of the revised manuscript). By using great care in the sample mounting, we are indeed able to reproduce our strain cell results with the piezostack method. We are therefore confident in the strain cell results presented in this manuscript, even when they differ from the results of Ishida *et al.*. The larger error bars at low temperatures in the piezo-stack data compared to the strain-cell data are due to the reduced strain sweep of the piezostack at low temperatures.

Fig. R3 Comparison of strain-cell-based (red triangles) and piezo-based (black squares) elastoresistance measurements for RbFe_2As_2 with strain applied along [100]. The strain-cell data have been scaled upwards by a factor of $(1/0.6)$ to account for the fact that $\epsilon_{true} = 0.6 \epsilon_{nom}$ in our strain cell. Note that the “strain cell” data here are the same data as the red filled squares in Fig. R1c.

2. As indicated in Fig. 1d, the authors measured longitudinal and transverse elastoresistance on the same sample. Note that they measured on two separated bar shaped samples in Ref. 45. While I think the longitudinal measurement is fine, the transverse measurement could have some problems. With stripes of conductive paints shortcut two edges, can the ‘transverse’ contacts pick up the real transverse resistance? I appreciate the authors for reporting their new set of data instead of taking their previous data of CsFe_2As_2 from Ref.45, that makes a comparison possible. The A_{1g} nematicity of CsFe_2As_2 reported in this new paper (Fig.5a) is 60% larger than that in Ref. 45 (Fig.2e), and the B_{2g} nematicity looks totally different. That raises the concern to what extent should we trust such results. Does the different way how contacts were made cause such mismatching?

We do not understand exactly what the referee means by “shortcut two edges.” We were careful to avoid electrical shorts between any two of the eight contacts. We believe that our contact configuration can pick up the true transverse elastoresistance as has been demonstrated in other works [M. He *et al.*, Nat. Commun. **8** 504 (2017)] and reproduced in our setup on BaFe_2As_2 (Supplementary Fig. S1 of the revised manuscript).

The referee is correct to point out the A_{1g} elastoresistance coefficient in our current paper is larger than our previous paper. This is not caused by the different contact configuration, but results from a difference in the ratio between real and nominal strain, as explained in detail below. This difference should have been explained in the manuscript. We have now added a corresponding statement in the Methods and apologize for the oversight.

In our original paper, we measured using a Razorbill Instruments CS-100 cell with only four contacts on the sample where the effective strained sample length was ~ 1 mm. In this work, in order to make room for 8 handmade electrical contacts, we increased the strained sample length to ~ 2 mm. When the sample length is longer, we require greater displacements of the sample mounting plates in order to compensate for the thermal contraction of the sample on cooling. For a sample length of 2mm, the required displacements exceeded the capabilities of the CS-100 cell. Therefore, the measurements in this paper were conducted instead in a Razorbill CS-120 cell, which has the capability to apply larger displacements. According to the official documentation provided with our strain cells by Razorbill Instruments, due to the deformation of the strain device itself by a stiff sample, the capacitance sensor will over-read the applied displacement according to

$$\frac{\Delta L_{meas}}{\Delta L_{real}} = 1 + \frac{k_s}{k_\tau} (h_s + h_a)(h_s + h_c)$$

where k_s is the sample spring constant, and h_s is the sample mounting height above the top of the apparatus. The remaining parameters are properties of the strain cell itself: k_τ is the torsional spring constant, h_a is the height of the top surface above the moving block rotation center and h_c is the height of the top surface above the center of the capacitor. The parameters k_τ , h_a and h_c are different between the two strain cells (precise information was provided only for the CS-100 cell). Therefore, due to different mechanical properties of the strain cells themselves, the relationship between measured capacitance and true applied strain may differ in the two strain cells. This ultimately affects the value of the measured elastoresistance. Furthermore, the sample spring constant k_s is inversely proportional to sample length. Typical sample spring constants in our original paper were ~ 4 N/ μ m ($L = 1$ mm) while typical spring constants in our current paper are ~ 2 N/ μ m ($L = 2$ mm). As is easily seen from the formula above, the smaller sample spring constant allows for a transmitted strain closer to the nominal value determined from the capacitance sensor, producing a greater measured elastoresistance in our current work. Due to these issues, we chose to re-measure CsFe₂As₂ in the CS-120 cell to make sure the results were comparable to our results for the other samples.

The referee also points out that the B_{2g} elastoresistance curve looks different from the previous work, primarily at low temperatures. In the previous work, the longitudinal and transverse were measured on different crystals. This brings up the issue of possibly different sample alignment and strain transmission between the longitudinal and transverse measurements. Additionally, uncertainty in the estimate of the unstrained sample length used in calculating the strain could differ between longitudinal and transverse measurements. Therefore, it was difficult to disentangle the much smaller B_{2g} coefficient from the larger A_{1g} coefficient and the error bars on the extracted elastoresistance coefficients were quite large. For these reasons, we consider the

present data on the temperature dependence of the B_{2g} coefficient to be more reliable. We note that there is agreement within the error bars presented in our previous work.

According to Ref. 1, the modified Montgomery contact is ideal for elasto-resistance measurements. I would like to suggest the authors to try this modified Montgomery configuration on at least one selected sample to verify their present results. Note that the Montgomery configuration was tested on one sample in Ref. 39, and their conclusion is that the A_{1g} contribution is negligibly small.

The Montgomery configuration requires a square-shaped sample which is incompatible with measurements in the strain cell, for which the sample must be rectangular. The edges of the sample submerged in epoxy are not strained. Therefore, our eight-contact configuration is the only possibility to measure longitudinal and transverse on the same experimental run.

We also point out that the sample measured via the Montgomery method by Ishida *et al.* is the $\text{Rb}_{0.65}\text{Ba}_{0.35}\text{Fe}_2\text{As}_2$ sample, which has very similar properties to our $\text{K}_{0.6}\text{Ba}_{0.4}\text{Fe}_2\text{As}_2$ sample. In this sample we also find a negligible A_{1g} component (Fig. 5a of the manuscript, downward pointing triangles). Ishida *et al.* have not conducted Montgomery configuration measurements on the pure KFe_2As_2 , RbFe_2As_2 and CsFe_2As_2 where we observe a large A_{1g} component.

3. The authors stated that they use DevCon 5 minutes Epoxy to mount their samples on the strain cell. The DevCon Epoxy is very soft. The effective strain transmitted to the sample was verified for gluing the sample on the surface of a piezo stack in Ref. 25. For Razorbill-type strain cells, a much stronger epoxy Stycast is frequently used. Stycast has been verified to be a good choice to transmit full strain to the sample (see, for example Nat Mater 16 708). As far as I know, such test has not been done for DevCon Epoxy with Razorbill cells. That makes one wonder whether the samples experience the strain applied to it. For example in Ref. 2a, and 2c, some of the black data points are surprisingly close to zero, which is strange. The deformation of a simple metal plate will result in an elasto-resistance of the order of 1. A measured elasto-resistance near 0 may imply the sample does not feel strain at all. I recommend the authors either demonstrate DevCon works as well as Stycast or simply try to use Stycast for at least one of their samples.

As detailed above, and now presented in the Supplemental Information (Supplementary Fig. S1 of the revised manuscript), the measurement of pure BaFe_2As_2 in our setup and comparison to the accepted values demonstrates that the sample in our setup feels 60% of the nominal strain determined from the capacitance sensor. This is similar to the 70% estimated elsewhere where stycast was used [A. Stern *et al.*, Nat. Mater. **16**, 708 (2017); C. W. Hicks *et al.*, Science **344**, 283 (2014); M. S. Ikeda *et al.*, Phys. Rev. B **98**, 245133 (2018)].

Further confidence in our strain transmission comes from the quantitative agreement between the strain dependence of the A coefficient of the resistance ($dA/d\varepsilon$) with the $d\gamma/d\varepsilon$ inferred from thermodynamic measurements in KFe_2As_2 (Supplemental Information). We also directly measured the strain-dependence of T_c in KFe_2As_2 , which is in reasonable agreement with the value from thermodynamics (Supplemental Information).

In our experience, stycast is soft at high temperature, whereas DevCon 5 minute epoxy has been demonstrated to give good strain transmission below ~ 260 K [J. C. Palmstrom *et al.*, Phys. Rev. B **96**, 205133 (2017)]. Since we want to measure elastoresistance over a large temperature range, we prefer the DevCon.

The referee notes that some of the black data points in Figs. 2a and 2c of the manuscript are close to zero and wonders whether or not the sample feels strain at all. First of all, we remind the referee that the dark yellow data points in the same panels with non-zero elastoresistance were obtained simultaneously. The small longitudinal elastoresistance for strain along [100] observed at high temperature in Figs. 2a and 2c of the manuscript results from a partial cancellation between the positive m_{A1g} and the negative m_{B1g} , as seen in Figs. 5a and 5c of the manuscript, and does not result from poor strain transmission. To prove that the low elastoresistance values do not arise simply from poor strain transmission on one experimental run, in Fig. R4, we show additional data from another experiment which are consistent with these data. The sample mounting details and procedure were exactly the same on all runs. These data demonstrate that our strain transmission is well reproducible between experiments.

Finally, it is important to point out that imperfect strain transmission does not affect our conclusions regarding the symmetry decomposition of the elastoresistance. Since our longitudinal and transverse elastoresistance measurements are conducted on the same sample on the same experimental run, imperfect strain transmission merely produces a scale factor on the measured elastoresistance, but does not affect the relative symmetry decomposition.

Fig. R4 Elastoconductance data on RbFe_2As_2 with strain applied on $[100]$ for two samples. The scattering of the longitudinal data on sample A may have resulted from poor mechanical stability of the electrical contacts.

Besides, some issues should be clarified before the paper is published

1. In the paragraph from line #69 to line #75, the authors refuted the attempt to associate a phase transition with the drops in the elastoconductance in Ref. 39. Based on the fact that no hint of phase transition is seen in $R(T)$ or $L(T)$ curves. In my opinion, it is a strong argument against a phase transition. However, I think the interpretation for such drop should be discussed in more details. Note that the elastoconductance of many samples with much lower RRR than the AFe_2As_2 samples have been reported in the literatures, but a drop in the elastoconductance is very rare. For example, the resistance of $\text{Ba}(\text{Fe}_{0.955}\text{Ni}_{0.045})_2\text{As}_2$ reported in Ref. 1 becomes comparable to its residual resistance at ~ 50 K (Fig. S11), but only a slight drop in elastoconductance is presented just above the T_c (~ 20 K) of this sample (Fig. 2B).

In the paragraph from line #69 to line #75 of the original submission, we refer to kinks in our own elastoconductance data at $T = 15$ K, not to the drop of elastoconductance reported by Ishida *et al.* ("Ref. 39") in a higher temperature range ($T = 38$ K).

As we have discussed in detail (Fig. 4 of the manuscript), these kinks in $m = [1/R]dR/d\varepsilon$ occur when the resistance R becomes comparable to the residual resistance R_0 . These kinks do not occur when the elastoconductance is instead defined as $\bar{m} = [1/(R - R_0)]dR/d\varepsilon$. Since the RRR of our samples is very high ($\text{RRR} \sim 1000$ in RbFe_2As_2), the residual resistance is completely negligible over most of the temperature range and $\bar{m} \approx m$ at high temperature, as seen in Fig. 4 of the manuscript. Only below 15 K is R sufficiently small that $\bar{m} \neq m$, leading to a kink in m .

Based on these arguments, the referee intuitively expects a kink in m when " R becomes comparable to R_0 ". The referee therefore wonders why no such kink is observed in

previously reported data on $\text{Ba}(\text{Fe}_{0.955}\text{Ni}_{0.045})_2\text{As}_2$, where $\text{RRR} \sim 2$ and therefore R is comparable to R_0 [H. Kuo *et al.*, Science **352** 958 (2016)].

The point is that, in $\text{Ba}(\text{Fe}_{0.955}\text{Ni}_{0.045})_2\text{As}_2$, the resistance R is *always* comparable to the residual resistance R_0 in the measured temperature range, so that $\bar{m} \neq m$ over the entire temperature range. This is illustrated in Fig. R5, where we have calculated \bar{m} from the existing data [H. Kuo *et al.*, Science **352** 958 (2016)] using an estimate for R_0 . Here, no kinks are visible in m , because there is no transition to a regime of negligible R_0 , as there is in RbFe_2As_2 .

Fig. R5 The $m_{B_{2g}}$ elastoresistance coefficient of $\text{Ba}(\text{Fe}_{0.955}\text{Ni}_{0.045})_2\text{As}_2$ [H. Kuo *et al.*, Science **352** 958 (2016)]. Using the freestanding resistance curves (inset), we calculate $\bar{m}_{B_{2g}}$ for two reasonable values of R_0 .

Wiechi *et al.* also mentioned the drop of elastoresistance can be a result of the coherence-incoherence crossover (They emphasized such idea in the paragraph from line #145 to line #149). However, in my opinion, it is not straightforward to accept this claim just based on their similar temperature range. The authors should explain how these two phenomena are related.

Our claim that the broad peak in the $m_{A_{1g}}$ coefficient of CsFe_2As_2 results from the coherence-incoherence crossover has a clear motivation, as discussed in the manuscript and supplement. This claim is not simply based on a similar temperature range.

The coherence-incoherence crossover in AFe_2As_2 is most clearly seen in the thermal expansion coefficient divided by temperature α/T [F. Hardy *et al.*, Phys. Rev. B **94**, 205113 (2016)]. α/T is a thermodynamic quantity that measures the derivative of the entropy with respect to symmetry-preserving stresses. This is related to the strain derivative of the entropy via elastic constants. Thus α/T is a measure of the A_{1g} strain dependence of the entropy. Meanwhile, the A_{1g} elastoresistance is the A_{1g} strain dependence of the resistance. Intuitively then, the A_{1g} elastoresistance should be

related to α/T to the extent that resistance is a good measure of the entropy. This appears to be the case in AFe_2As_2 , since the coherence temperature is clearly seen in resistance. This intuitive argument is made more solid by an application of the Fisher-Langer relation, which reveals a clear correspondence in the behavior of m_{A1g} and α/T in AFe_2As_2 (see section "Qualitative comparison between elastoresistance and thermal expansion" and Fig. S4 in the Supplemental Information).

Ishida *et al.* observed a broad peak in the longitudinal elastoresistance of RbFe_2As_2 with strain applied along [100], which they attributed to a B_{1g} nematic phase transition. We do see a similar maximum (black curve Fig. 2c of the manuscript). However, a B_{1g} nematic phase transition would be associated with a peak in the symmetry-resolved elastoresistance coefficient m_{B1g} . Despite the peak in [100] longitudinal elastoresistance, we find no peak in the symmetry-resolved m_{B1g} coefficient (Fig. 5c of the manuscript).

2. I do not think it is reasonable to treat AFe_2As_2 and $\text{K}_{1-x}\text{Ba}_x\text{Fe}_2\text{As}_2$ as a continuous doping series (Fig. 6 and the related discussion parts in the main text). That is because (i) Substituting K to Ba is hole doping, while AFe_2As_2 is isovalent doping (ii) The lattice parameter a decreases from BaFe_2As_2 to KFe_2As_2 , while it increases from KFe_2As_2 to CsFe_2As_2 . If the authors wish to show some decent doping dependence, I think they should measure more samples of $\text{K}_{1-x}\text{Ba}_x\text{Fe}_2\text{As}_2$ with different x , and some samples like $\text{K}_{1-x}\text{Cs}_x\text{Fe}_2\text{As}_2$.

3. In vein of my previous comment, the 'monotonic' increase of $A1g$ response and the 'weakening' of $B2g$ response with hole doping as the authors stated in the introduction part (line #44) is only based on two points, $x=0.6$ and $x=1.0$. I do not think any trend can be summarized based on two points. Furthermore, if one includes the $x=0.4$ case reported in Ref. 1, he will find that the $B2g$ response is nearly the same as $x=0.6$ reported in this paper. So, there is no 'monotonic' trend. This raises another question, why $B2g$ is not reduced from $x=0.4$ to $x=0.6$? It is accepted that nematic fluctuation is strong at the critical point ($x=0.4$), and should decrease when it is away from the criticality. However, it seems not to be the case in $\text{K}_{1-x}\text{Ba}_x\text{Fe}_2\text{As}_2$.

While we agree that Fig. 6 of the manuscript is not truly a doping phase diagram, it is a convenient way of summarizing the data in terms of a trend of increasing γ . This style of presentation has been used elsewhere [F. Hardy *et al.*, Phys. Rev. B **94**, 205113 (2016)] and should not cause confusion.

4. Additionally, it is far-fetched to claim that the extended superconducting dome in $\text{K}_{1-x}\text{Ba}_x\text{Fe}_2\text{As}_2$ is related to the remnants of the $B2g$ nematicity, as the authors wrote from line #166. Nematicity is expected to enhance superconductivity only for certain types of pairing. The superconducting states of hole-doped $\text{K}_{1-x}\text{Ba}_x\text{Fe}_2\text{As}_2$ is very complicated. It is likely to change from S_{+-} to s_{+i} to nodal (probably d -wave) with increasing x . There is even a Lifshitz transition around $x=0.8$, which the authors also pointed out. It is hard to believe the $B2g$ nematicity can be a

determinative factor to T_c in such a complicated situation. It is even no consensus whether nematicity matters at all in any of these superconducting states.

We agree with the referee that this claim is more of a speculation. Therefore, we have deleted the statement: "This may be related to why the superconducting dome stretches so far in these compounds" on line #168 of the original submission.

5. In the paragraph from line #84 to line #90, the authors pointed out that the tensile strain has the same effect as the negative chemical pressure (from $A = K$ to Cs) on the parameter A . However, as they presented, T_c of KFe_2As_2 increases under tensile strain, while decreasing with negative chemical pressure. In that sense, is the same trend presented by parameter A just a coincidence?

The referee is correct. For the A coefficient of resistance, A increases with both physical tensile strain and negative chemical pressure ($K \rightarrow Rb \rightarrow Cs$). In contrast T_c increases with physical tensile strain, but decreases with negative chemical pressure ($K \rightarrow Rb \rightarrow Cs$). Therefore, the analogy between physical and chemical pressure breaks down for the superconducting transition. This is not surprising, since the superconducting T_c depends on many different factors.

We have modified the manuscript and now refer only to the consistency with thermodynamic quantities, without reference to the unnecessary chemical pressure analogy. We therefore remove the sentence (line 85-88): "The increase of A under tensile strain ($\epsilon > 0$) is also consistent with the increase of the Sommerfeld coefficient in unstrained AFe_2As_2 ($A = K, Rb, Cs$) (Fig. 1a) from $A = K$ to $A = Cs$, since the increasing alkali ion size from K to Cs creates a negative chemical pressure."

6. The authors mentioned in line #199 that DuPont 4929N sliver paint were found to be mechanically unstable on strain application. I think it would be helpful if the authors can provide more details in the supplementary materials. As far as I know, DuPont 4929N is wildly used in transport measurements, including many of the previous elastoresistance studies. Does it mean all the results of the previous publication involved contacts made with DuPont 4929N need to be revisited?

We certainly do not discount previous work on the basis that DuPont 4929N was used. The problem may be specific to the air-sensitive surfaces of AFe_2As_2 samples. Similar difficulties were noted for measurements of KFe_2As_2 by other groups, who resorted to soldering wires onto the sample [Taufour *et al.*, Phys. Rev. B **89**, 220509(R) (2014) (Supplementary Information); X.C. Hong and C. Hess (IFW Dresden), private communication].

There is no need to revisit previous results because there is in principle no difference between resistance measurements made with DuPont 4929N and measurements made

with our Hans Wolbring Leitsilber 200N provided that the contacts are stable and the contact resistances are on the order of 2 Ohms.

7. In the paragraphs from line #91 to line #106, the authors redefined the elastoresistance m and argued it produces more physically meaningful results. I think it is an interesting finding. The origin of the elastoresistance can be anisotropic scattering rate or Pomeranchuk instability of the Fermi surface. As far as I know, there is no conclusion on the leading factor of nematicity in iron-based superconductors yet. The analysis of this paper seems to be a strong support of the scattering scenario. Could the authors discuss in more detail on this point? Again, does it mean we should re-analysis all the published elastoresistance results to get this redefined m , in order to get more physically meaningful results? Note that the RRR values of other iron-based superconductors are much smaller than AFe_2As_2 . So, the calculated elastoresistance of other iron-based superconductors will change drastically if R_0 is considered.

Indeed, as we show Fig. R5 of this response, the temperature dependences of \bar{m} and m are quite different for many materials of interest. Perhaps this could shed some light on the unexplained "sub-Curie Weiss" behavior observed in many iron-based materials, which has been explained in terms of disorder. While we are exploring such interesting possibilities, a full discussion of these points clearly lies outside the scope of the current work.

The referee says that "origin of the elastoresistance can be anisotropic scattering rate or Pomeranchuk instability of the Fermi surface." This is true for B_{1g} or B_{2g} elastoresistance. However, one of the key points of our work is that elastoresistance can arise from sources other than nematicity. In this work, we have demonstrated that in AFe_2As_2 , strain can modulate the isotropic scattering rate, leading to A_{1g} elastoresistance.

To conclude, I do not recommend this paper to appear on Nature Communication in its current form.

We thank the referee for their careful reading of the manuscript. We hope that our responses have eased their concerns. Since this response has been rather long, we would like to summarize our responses by emphasizing that, regardless of any discrepancies with the raw data of Ishida *et al.*, the conclusions drawn from our data are supported by their good agreement and consistency with well-established thermodynamic measurements on these materials. Finally, we remind the referee that the data presented by Ishida *et al.* are clearly insufficient to claim B_{1g}/XY nematicity, as they lack the proper symmetry decomposition.

REVIEWERS' COMMENTS

Reviewer #1 (Remarks to the Author):

The authors have addressed all my concerns, the paper is now suitable for publication.

Reviewer #2 (Remarks to the Author):

This is my second report on this manuscript. The authors have provided reasonable responses to many of the referees' points, including mine. Unfortunately, direct proofs that the data of this work instead of Ref. 39 is correct, which I asked in previous report, are not presented in the second version of the manuscript and supplementary materials. In the Rebut Letter, the authors claimed the mismatch between the strain-cell and piezoelectric-stack results is due to the sample damage of the strain-cell and piezoelectric-stack method. They stressed that "it is more plausible that a damaged sample no longer responds purely elastically to strain." However, as they pointed out, their piezoelectric-stack results of CsFe₂As₂ (shown in their previous paper Ref. 45) agrees quite well with Ref. 39. What a coincidence it should be to let two damaged samples to get the similar (wrong) results? In my opinion, this piece of information is rather a proof that the results of Ref. 39 cannot be simply attributed to sample damage. The authors argued "We note that such a large jump of the longitudinal elastoresistance between 86% and 100% Rb doping may be implausible." However, such results of Ref. 39 were supported by careful theoretic consideration, for example PRL 123, 146402 and PRB 100, 020507. It is not a serious judgement that the results of Ref. 39 is implausible just because the jump is large.

Nevertheless, the authors indeed provided Fig. R3, which demonstrated if the RbFe₂As₂ sample was mounted "using great care", perfect agreement between strain-cell and piezoelectric-stack results is achieved, a direct proof that Ref. 39 is totally wrong at the raw data level. Thus its following theoretical works are not relevant to real-world materials. This is exactly the important information that can "revise the current understanding of this topic". Although I still cannot believe why a proper/improper sample mounting can let the results agree very well with damaged/ undamaged samples respectively, it is the message that all readers of this paper should know. So I believe Fig. R3 should not stay in the Rebut Letter, which most readers will not read (or even do not have access to). Instead, it should be emphasized in the main text as the central finding of this work. It will be nice if the author can provide similar results also for CsFe₂As₂. I will recommend this paper accepted as a Nature Communications if the authors accept this advice.

Besides, I still believe Fig. 6 is quite misleading. The authors argued "it is a convenient way of summarizing the data in terms of a trend of increasing gamma. This style of presentation has been used elsewhere [F. Hardy et al., Phys. Rev. B 94, 205113 (2016)] and should not cause confusion." The quoted PRB paper is from the same group. I do not think that proves such style of presentation is commonly accepted. Additionally, Fig. 3b of this PRB paper contains 4 points in the doped case, while this paper just has 1 point. Not to say the PRB paper used different colors to separate the "diagram". If the authors really want to show their data points in a unified diagram, I think the way Fig. 4 of Annu. Rev. Condens. Matter Phys. 5, 113 do is a much better.

Reviewer #3 (Remarks to the Author):

The manuscript studied the elastoresistivity in AFe₂As₂ (A=K,Rb,Cs) and concluded that no nematic order exists in these compounds. This conclusion disagrees with some other works that claimed there is a B1g nematic order. Having the privileges to read previous referees' reports, I recommend the

publication of this work to Nature Communications. The main reasons are listed as follows.

First, whether there are B_{1g} nematic fluctuations or even B_{1g} nematic order is a very important subject in iron-based superconductors. The appearance of B_{1g} nematicity may alter our view of the origin of superconductivity. This work gives solid studies on this issue from the elastoresistivity measurements and from my point of view, puts an end to the debates.

Second, while the current work is somehow related to Ref. [45] as pointed out by referee 2, I think it contains new information that is good enough to publish in Nature Communications. As the authors replied, they have successfully disentangled the elastoresistivity in all symmetry channels that are interested, which is new for the AFe₂As₂ systems. The detailed analysis will benefit the community for iron-based superconductors and strongly correlated electron systems to avoid recklessly attributing any kind of elastoresistivity behavior to nematicity. Moreover, personally, I would like to encourage people to publish results that may sound "negative" to some fancy concepts, especially when these concepts may be misleading or even wrong.

Third, although their results are different from Ref. [39], the analysis in the manuscript and detailed replies to referee 2 show that their data are more reliable. Especially, I would like to point it out that the measurements in Ref. [39] were done by gluing the crystals onto the piezoelectric stacks, which would result in local inhomogeneous stress. On the other hand, the studies here were carried out on freestanding samples, which should be more reliable. As shown by their replies to referee 2, the authors have also considered many factors that affect the results. Moreover, as pointed it out by the authors, Ref. [39] did not consider the contributions from different symmetry channels, which is insufficient to claim the XY or B_{1g} nematicity. Therefore, compared to the results of Ref. [39], I trusted those of the current work. It may be worth noting that our group has also measured the elastoresistivity of several RbFe₂As₂ single crystals in the freestanding way by using a uniaxial-pressure device that is different from that of the current work, we found that no kink that supports a nematic order (not published), consistent with the results reported in this manuscript.

While both referees have raised many questions on the technique issues and data analysis, I think the authors have well addressed them. So I don't have any further questions except for one minor issue. The authors stated that they have deleted the sentence "This may be related to why the superconducting dome stretches so far in these compounds", but I still find it in line 170.

Reviewer #1 (Remarks to the Author):

The authors have addressed all my concerns, the paper is now suitable for publication.

We thank the referee for their recommendation to publish our manuscript in Nature Communications.

Reviewer #2 (Remarks to the Author):

This is my second report on this manuscript. The authors have provided reasonable responses to many of the referees' points, including mine. Unfortunately, direct proofs that the data of this work instead of Ref. 39 is correct, which I asked in previous report, are not presented in the second version of the manuscript and supplementary materials. In the Rebut Letter, the authors claimed the mismatch between the strain-cell and piezoelectric-stack results is due to the sample damage of the strain-cell and piezoelectric-stack method. They stressed that "it is more plausible that a damaged sample no longer responds purely elastically to strain." However, as they pointed out, their piezoelectric-stack results of CsFe₂As₂ (shown in their previous paper Ref. 45) agrees quite well with Ref. 39. What a coincidence it should be to let two damaged samples to get the similar (wrong) results? In my opinion, this piece of information is rather a proof that the results of Ref. 39 cannot be simply attributed to sample damage. The authors argued "We note that such a large jump of the longitudinal elastoresistance between 86% and 100% Rb doping may be implausible." However, such results of Ref. 39 were supported by careful theoretic consideration, for example PRL 123, 146402 and PRB 100, 020507. It is not a serious judgement that the results of Ref. 39 is implausible just because the jump is large. Nevertheless, the authors indeed provided Fig. R3, which demonstrated if the RbFe₂As₂ sample was mounted "using great care", perfect agreement between strain-cell and piezoelectric-stack results is achieved, a direct proof that Ref. 39 is totally wrong at the raw data level. Thus its following theoretical works are not relevant to real-world materials. This is exactly the important information that can "revise the current understanding of this topic". Although I still cannot believe why a proper/improper sample mounting can let the results agree very well with damaged/ undamaged samples respectively, it is the message that all readers of this paper should know. So I believe Fig. R3 should not stay in the Rebut Letter, which most readers will not read (or even do not have access to). Instead, it should be emphasized in the main text as the central finding of this work. It will be nice if the author can provide similar results also for CsFe₂As₂. I will recommend this paper accepted as a Nature Communications if the authors accept this advice.

We agree with the referee that the agreement between the strain-cell-based and piezoelectric-stack-based measurements in RbFe_2As_2 is important for readers to know. We have now included this data as an inset to Fig. 2 of the manuscript.

Besides, I still believe Fig. 6 is quite misleading. The authors argued "it is a convenient way of summarizing the data in terms of a trend of increasing gamma. This style of presentation has been used elsewhere [F. Hardy et al., Phys. Rev. B 94, 205113 (2016)] and should not cause confusion." The quoted PRB paper is from the same group. I do not think that proves such style of presentation is commonly accepted. Additionally, Fig. 3b of this PRB paper contains 4 points in the doped case, while this paper just has 1 point. Not to say the PRB paper used different colors to separate the "diagram". If the authors really want to show their data points in a unified diagram, I think the way Fig. 4 of Annu. Rev. Condens. Matter Phys. 5, 113 do is a much better.

In order to avoid reader confusion, we have modified the axis color in Fig. 6 to emphasize the difference between the K-doping series and the K, Rb, Cs chemical substitution.

Reviewer #3 (Remarks to the Author):

The manuscript studied the elastoresistivity in AFe_2As_2 ($A=\text{K,Rb,Cs}$) and concluded that no nematic order exists in these compounds. This conclusion disagrees with some other works that claimed there is a B_{1g} nematic order. Having the privileges to read previous referees' reports, I recommend the publication of this work to Nature Communications. The main reasons are listed as follows.

First, whether there are B_{1g} nematic fluctuations or even B_{1g} nematic order is a very important subject in iron-based superconductors. The appearance of B_{1g} nematicity may alter our view of the origin of superconductivity. This work gives solid studies on this issue from the elastoresistivity measurements and from my point of view, puts an end to the debates.

Second, while the current work is somehow related to Ref. [45] as pointed out by referee 2, I think it contains new information that is good enough to publish in Nature Communications. As the authors replied, they have successfully disentangled the elastoresistivity in all symmetry channels that are interested, which is new for the AFe_2As_2 systems. The detailed analysis will benefit the community for iron-based superconductors and strongly correlated electron systems to avoid recklessly attributing any kind of elastoresistivity behavior to nematicity. Moreover, personally, I would like to encourage people to publish results that may sound "negative" to some fancy concepts, especially when these concepts may be misleading or even wrong.

Third, although their results are different from Ref. [39], the analysis in the manuscript and detailed replies to referee 2 show that their data are more reliable. Especially, I would like to point it out that the measurements in Ref. [39] were done by gluing the

crystals onto the piezoelectric stacks, which would result in local inhomogeneous stress. On the other hand, the studies here were carried out on freestanding samples, which should be more reliable. As shown by their replies to referee 2, the authors have also considered many factors that affect the results. Moreover, as pointed it out by the authors, Ref. [39] did not consider the contributions from different symmetry channels, which is in sufficient to claim the XY or B1g nematicity. Therefore, compared to the results of Ref. [39], I trusted those of the current work. It may be worth noting that our group has also measured the elastoresistivity of several RbFe₂As₂ single crystals in the freestanding way by using a uniaxial-pressure device that is different from that of the current work, we found that no kink that supports a nematic order (not published), consistent with the results reported in this manuscript.

While both referees have raised many questions on the technique issues and data analysis, I think the authors have well addressed them. So I don't have any further questions except for one minor issue. The authors stated that they have deleted the sentence "This may be related to why the superconducting dome stretches so far in these compounds", but I still find it in line 170.

We thank the referee for their recommendation to publish our manuscript in Nature Communications. We also thank the referee for pointing out the oversight, which has been corrected.